# Molecular mechanism of Afadin substrate recruitment to the receptor phosphatase PTPRK via its pseudophosphatase domain

Iain M Hay[1,2], Katie E Mulholland[2], Tiffany Lai[2], Stephen C Graham[3], Hayley J Sharpe[2]*, Janet E Deane[1]*

[1]Cambridge Institute for Medical Research, University of Cambridge, Cambridge, United Kingdom; [2]Signalling Programme, Babraham Institute, Cambridge, United Kingdom; [3]Department of Pathology, University of Cambridge, Cambridge, United Kingdom

**Abstract** Protein tyrosine phosphatase receptor-type kappa (PTPRK) is a transmembrane receptor that links extracellular homophilic interactions to intracellular catalytic activity. Previously we showed that PTPRK promotes cell–cell adhesion by selectively dephosphorylating several cell junction regulators including the protein Afadin (Fearnley et al, 2019). Here, we demonstrate that Afadin is recruited for dephosphorylation by directly binding to the PTPRK D2 pseudophosphatase domain. We mapped this interaction to a putative coiled coil (CC) domain in Afadin that is separated by more than 100 amino acids from the substrate pTyr residue. We identify the residues that define PTP specificity, explaining how Afadin is selectively dephosphorylated by PTPRK yet not by the closely related receptor tyrosine phosphatase PTPRM. Our work demonstrates that PTP substrate specificity can be determined by protein–protein interactions distal to the active site. This explains how PTPRK and other PTPs achieve substrate specificity despite a lack of specific sequence context at the substrate pTyr. Furthermore, by demonstrating that these interactions are phosphorylation-independent and mediated via binding to a non-catalytic domain, we highlight how receptor PTPs could function as intracellular scaffolds in addition to catalyzing protein dephosphorylation.

*For correspondence:
hayley.sharpe@babraham.ac.uk
(HJS);
jed55@cam.ac.uk (JED)

**Competing interest:** The authors declare that no competing interests exist.

## Editor's evaluation

This Research Advance follows up on the authors' recent *eLife* article where they reported the identification of a series of pTyr protein targets for the protein–tyrosine phosphatase (PTP) activity of the receptor-like PTP, PTPRK. Here they identify a docking site in the catalytically inactive D2 pseudophosphatase domain that promotes substrate dephosphorylation by the D1 phosphatase domain. The evidence for the docking interaction is convincing, and the study is important because it suggests that the D2 pseudophosphatase domains of other RPTPs may similarly function in substrate recruitment and selectivity. The article will be of interest to biochemists studying protein phosphorylation–dephosphorylation and signal transduction mechanisms.

## Introduction

Tyrosine phosphorylation is a key post-translational modification in cellular communication, enabling cells to dynamically adapt their behavior in response to external cues. Kinases (PTKs) and phosphatases (PTPs) act in concert to maintain cellular phosphotyrosine (pTyr) levels. Dysregulation of this balance is associated with disease, including tumor progression (*Arora and Scholar, 2005*; *Hunter and Sefton, 1980*). The importance of these pathways is exemplified by the clinical use of PTK inhibitors and

efforts to therapeutically target PTPs. It has been challenging to develop PTP inhibitors due to their highly charged active sites, as well as their lack of selectivity at the peptide level (*Barr et al., 2009*). Despite a lack of specific consensus sequences for pTyr substrates (*Barr et al., 2009*; *Ren et al., 2011*), PTPs show marked substrate specificity in vivo (*Fearnley et al., 2019*; *Saxena et al., 1999*). Thus, understanding how PTPs engage their substrates could lead to the development of alternative strategies to target them in the clinic.

Protein tyrosine phosphatase receptor-type kappa (PTPRK) is a member of the R2B family of transmembrane receptor (R)PTPs that supports cell adhesion and has been implicated as a tumor suppressor (*Chang et al., 2020*; *McArdle et al., 2001*; *Shimozato et al., 2015*). Like other receptor PTPs, PTPRK possesses two intracellular PTP domains, one catalytically active proximal to the plasma membrane (D1) and a distal inactive, pseudophosphatase (D2). We previously showed that for several PTPRK substrates the active D1 PTP domain itself appears not to be the sole determinant of substrate binding. Instead, the inactive D2 domains of PTPRK and its paralog PTPRM were shown to have critical roles in substrate recruitment (*Fearnley et al., 2019*). Specifically, we identified Afadin (MLLT4/AF6) as a PTPRK substrate and identified that it is recruited via the PTPRK D2 domain (*Fearnley et al., 2019*). In a similar paradigm, structural studies of the PTPRC-ICD suggested a potential role for the D2 domain in substrate recruitment (*Nam et al., 2005*). Indeed, it has been shown that the presence of the PTPRC-D2 is critical for the recruitment and dephosphorylation of several proposed substrates, including the T cell receptor (*Kashio et al., 1998*) and the Lck Src-family tyrosine kinase (*Felberg et al., 2004*).

Afadin is a conserved junctional plaque protein, important for linking cell surface adhesion molecules to the cytoskeleton, and is regulated by tyrosine phosphorylation (*Niessen and Gottardi, 2008*; *Yu and Zallen, 2020*). Afadin deletion in mice is embryonically lethal and leads to dysregulated junctions and polarity (*Zhadanov et al., 1999*). It is a large (>200 kDa) protein possessing multiple folded domains and extended regions of predicted disorder. The annotation of these domains, as described in UniProt (UniProt ID: P55196), is in good agreement with predictions of ordered/structured regions as calculated by AlphaFold2 (AF2) and IUPred3 (*Erdős et al., 2021*; *Jumper et al., 2021*; *Figure 1A*, *Figure 1—figure supplement 1*). This modular domain structure allows Afadin to engage in multivalent interactions through distinct domains. Afadin is an effector of GTP-bound Ras (*Goudreault et al., 2022*; *Smith et al., 2017*), placing it downstream of activated receptor tyrosine kinases. It can also play a structural role, linking adhesion receptors, such as nectins, to the actin cytoskeleton (*Takai and Nakanishi, 2003*). It participates in additional scaffolding functions, for example, its PDZ domain binds the plekstrin homology (PH) domain-containing adherens junction protein PLEKHA7 (*Kurita et al., 2013*). Its interaction with LGN, a protein involved in spindle orientation during cytokinesis, was characterized structurally and showed binding to a C-terminal disordered region of Afadin, adjacent to its F-actin binding region (*Carminati et al., 2016*). This structure clearly demonstrates that these disordered regions of Afadin can engage in specific interactions via formation of elongated peptide interfaces along extended surfaces of partner proteins. Within one of these extended, disordered regions of Afadin lies residue Y1230, the tyrosine that was identified as the putative target for dephosphorylation by the PTPRK D1 domain as it was hyperphosphorylated in PTPRK knockout (KO) cells (*Fearnley et al., 2019*). Phosphorylation of Afadin is required for its interaction with SHP-2 and is important for downstream signaling by platelet-derived growth factor (*Nakata et al., 2007*). Thus, Afadin is an essential regulator of adherens junctions that mediates phosphorylation-dependent and multivalent protein–protein interactions.

Using whole-cell lysates, we previously reported that the interaction between Afadin and PTPRK-D2 was independent of phosphorylation. This raises the question of whether the interaction is mediated by a region of Afadin that does not encompass the phospho-peptide or is mediated via an additional binding partner. Interestingly, the paralogous PTPRM did not recruit and dephosphorylate Afadin, indicating distinct specificities even within the same family of receptors. In contrast, the D2 domain of an inactive paralog PTPRU was sufficient to recruit Afadin to the PTPRK D1 for dephosphorylation (*Hay et al., 2020*). To define the molecular details of R2B PTP substrate specificity, the interaction of PTPRK with its substrate Afadin was investigated.

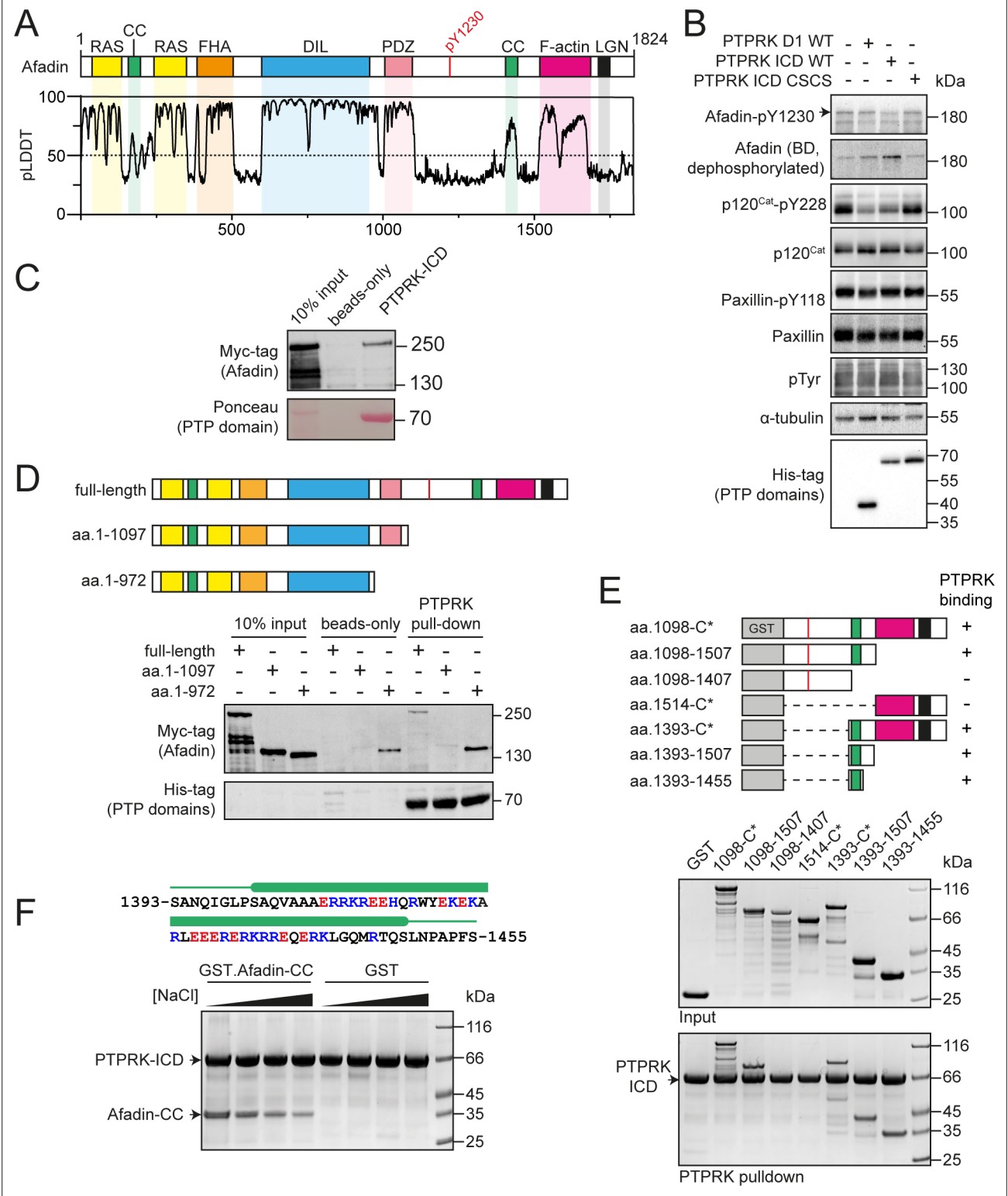

**Figure 1.** PTPRK interacts directly with the Afadin C-terminal coiled-coil (CC) region. (**A**) Top: schematic of full-length human Afadin, with domain annotation based on UniProt ID P55196, showing Ras-association (RAS, yellow), CC (green), forkhead-associated (FHA, orange), dilute (DIL, blue), PDZ (pink), F-actin binding (magenta) domains, and LGN-binding peptide motif (black). The putative PTPRK target site Y1230 is also highlighted (red). Bottom: predicted local distance difference test (pLDDT) for the Afadin AlphaFold2 (AF2) prediction from the AlphaFold Protein Structure Database (P55196, retrieved 7/2/2022). A pLDDT score <50 is predictive of protein disorder (*Jumper et al., 2021*). (**B**) Immunoblot analysis of pervanadate-treated MCF10A cell lysates incubated for 45 min at 4°C with 0.3 μM of the indicated recombinant PTP domains. Note: the total Afadin antibody is sensitive

*Figure 1 continued on next page*

*Figure 1 continued*

to phosphorylation likely at Y1230 (antigen: Afadin 1091–1233) and therefore indicates dephosphorylated Afadin. A full time course is quantified in *Figure 1—figure supplement 2E*. (**C**) Immunoblot analysis of streptavidin bead-conjugated PTPRK-ICD pull-downs from wheat germ lysate containing full-length Afadin. (**D**) Top: schematic of different Afadin C-terminal truncations used in initial mapping experiments, colored as in (**A**). Bottom: immunoblot analysis of streptavidin bead-conjugated PTPRK-ICD pull-downs from wheat germ lysates containing C-terminal Afadin truncations. Prey proteins enriched on both beads-only and PTPRK pull-downs were considered to be nonspecific interactions. (**E**) Top: schematic of different Afadin C-terminal GST-fusion constructs used for interaction mapping. Bottom: pull-downs using streptavidin bead-conjugated PTPRK-ICD with purified GST-Afadin fusion proteins, followed by SDS-PAGE and Coomassie staining. (**F**) Top: sequence of Afadin-CC showing predicted helical region (green block) as observed in the full-length Afadin AF2 prediction. This region contains a high number of charged residues, which have been highlighted (basic, blue; acidic, red). Bottom: pull-downs using streptavidin-conjugated-PTPRK-ICDs with either GST or GST-Afadin-CC in the presence of increasing NaCl concentrations (10, 100, 250, 500 mM; left to right) followed by SDS-PAGE and Coomassie staining. Gels and blots shown in this figure are representative of n ≥ 3 independent experiments.

The online version of this article includes the following source data and figure supplement(s) for figure 1:

**Source data 1.** Uncropped, unedited blots for *Figure 1B*.

**Source data 2.** Uncropped, unedited blots for *Figure 1C*.

**Source data 3.** Uncropped, unedited blots for *Figure 1D*.

**Source data 4.** Uncropped, unedited gels for *Figure 1E*.

**Source data 5.** Uncropped, unedited gels for *Figure 1F*.

**Figure supplement 1.** Disorder predictions for Afadin.

**Figure supplement 2.** Afadin-pY1230 antibody validation.

**Figure supplement 2—source data 1.** Uncropped, unedited blots for *Figure 1—figure supplement 2B*.

**Figure supplement 2—source data 2.** Uncropped, unedited blots for *Figure 1—figure supplement 2C*.

**Figure supplement 2—source data 3.** Uncropped, unedited blots for *Figure 1—figure supplement 2D*.

**Figure supplement 2—source data 4.** Densitometric quantification shown in *Figure 1—figure supplement 2E*.

**Figure supplement 2—source data 5.** Uncropped, unedited blots for *Figure 1—figure supplement 2F*.

**Figure supplement 2—source data 6.** Densitometric quantification shown in *Figure 1—figure supplement 2G*.

**Figure supplement 3.** Purification of in vivo biotinylated PTP domains.

**Figure supplement 3—source data 1.** Uncropped, unedited gels for *Figure 1—figure supplement 3A*.

**Figure supplement 3—source data 2.** Uncropped, unedited blots for *Figure 1—figure supplement 3B*.

## Results

### PTPRK dephosphorylates Afadin Y1230 in a D2 domain-dependent manner

Previous work has shown that the Afadin residue Y1230 is hyperphosphorylated in PTPRK KO epithelial cells, suggesting that this is a tyrosine targeted for dephosphorylation by PTPRK (*Fearnley et al., 2019*). To test this, we generated an antibody against a synthetic tyrosine phosphorylated peptide corresponding to amino acids 1224–1235 of Afadin (*Figure 1—figure supplement 2A*). Consistent with our previous tyrosine phosphoproteomics data, a band at the predicted molecular weight of the longest isoform of Afadin is detected at higher levels in PTPRK KO compared to WT MCF10A cells (*Figure 1—figure supplement 2B*). This band is increased by pervanadate (*Huyer et al., 1997*), a cell treatment that globally increases protein tyrosine phosphorylation, and is decreased by knockdown of Afadin using siRNA (*Figure 1—figure supplement 2C*). It should be noted that phosphorylation appears to interfere with the signal when using an anti-Afadin (epitope: 1091–1233) antibody from BD Transduction Laboratories on immunoblots, resulting in decreased intensity when Afadin is phosphorylated (*Figure 1—figure supplement 2A and C*). To further validate the site specificity of the Afadin-pY1230 antibody, we generated tyrosine to phenylalanine (YF) mutants of mScarlet-tagged Afadin. The antigenic phosphopeptide used as an epitope possesses two tyrosine residues, Y1226 and Y1230, which we mutated alone or in combination. mScarlet-Afadin phosphorylation, as detected by our antibody, increased with pervanadate treatment and was completely reversed by calf intestinal phosphatase treatment (*Figure 1—figure supplement 2D*). However, there was no detectable increase in signal upon pervanadate treatment of mScarlet-Afadin-Y1230F mutants, supporting that this is the predominant phosphosite recognized by our Afadin phosphoantibody.

We previously demonstrated that the PTPRK ICD, but not the PTPRK D1 domain alone, dephosphorylates Afadin (*Fearnley et al., 2019*). This is recapitulated using the pY1230 Afadin antibody after incubation of recombinant domains with pervanadate-treated, DTT-quenched, MCF10A lysates (*Figure 1B*, *Figure 1—figure supplement 2E*). Over a time course, PTPRK-ICD shows greater activity against Afadin pY1230 than PTPRK-D1. In contrast, p120^Cat was dephosphorylated equally by the two domains, with minimal dephosphorylation of the non-substrate control, paxillin-pY118. Finally, the observed hyperphosphorylation of Afadin Y1230 in PTPRK KO MCF10A cell lysates could be reduced by doxycycline-induced expression of PTPRK to wildtype levels, but not by a catalytically inactive mutant (*Figure 1—figure supplement 2F and G*). Together, these data confirm that this new antibody is specific to Afadin phosphorylated at Y1230 and that pY1230 is targeted for dephosphorylation by PTPRK in a D2 domain-dependent manner. This pTyr residue is located in the C-terminal half of Afadin within an extended region of predicted disorder (*Figure 1A*). We next sought to determine how Afadin is recognized by the PTPRK D2.

## Afadin binds directly to the PTPRK intracellular domain

The large size and predicted disorder of Afadin make it poorly suited to recombinant expression in *Escherichia coli*. Therefore, to assay direct binding of Afadin to PTPRK, recombinant full-length Afadin was expressed in vitro using a wheat germ-based cell-free translation system. Bacterially expressed and biotinylated PTPRK intracellular domain (ICD, *Figure 1—figure supplement 3*) was immobilized on streptavidin beads for pull-downs from wheat germ lysate containing full-length Afadin. We observed clear binding of full-length Afadin to the PTPRK-ICD (*Figure 1C*), consistent with a direct interaction. To determine whether PTPRK binds a region of Afadin incorporating tyrosine 1230, pull-downs were performed using C-terminal truncations of Afadin produced in wheat germ lysate. Deletion of the Afadin C-terminal disordered tail was sufficient to abolish binding to PTPRK (*Figure 1D*, aa. 1–1097). However, additional truncations showed substantial nonspecific binding to the streptavidin resin hindering further interaction mapping using this strategy (e.g., aa. 1–972, *Figure 1D*).

We next mapped PTPRK binding to the Afadin C-terminus in more detail using GST-fusion proteins (*Figure 1E*). A GST-fusion construct encompassing the entire C-terminal disordered region of Afadin, shown to be important in the wheat germ system, also showed direct binding to PTPRK, supporting the identification of this region as sufficient for binding (*Figure 1E*). Several truncation constructs within this region were tested for binding to PTPRK, clearly identifying that only constructs containing a short, 63-residue predicted coiled-coil (CC) region spanning residues 1393–1455 was able to bind PTPRK (*Figure 1E*). Therefore, this CC region is both necessary and sufficient for binding to the PTPRK-ICD. The sequence composition of this region of Afadin (Afadin-CC) includes a high proportion of charged residues (*Figure 1F*), suggesting that the PTPRK–Afadin interaction could be electrostatically mediated. In support of this, the PTPRK–Afadin interaction was sensitive to salt concentration, with less GST-Afadin-CC present in PTPRK pull-downs at increasing NaCl concentrations (*Figure 1F*). Further, PTPRK-ICD and Afadin-CC have complementary predicted isoelectric points (pI = 5.3 and 10.2, respectively), suggesting that these fragments would retain charge complementarity at physiological pH. Together, these data show that the PTPRK-ICD interacts directly and specifically with the C-terminal Afadin-CC region.

## Afadin-CC and PTPRK D2 form a 1:1 complex

CC domains consist of heptad repeats (designated a–g), in which residues 'a' and 'd' are generally nonpolar, forming a hydrophobic seam that facilitates oligomerization (*Truebestein and Leonard, 2016*). The primary sequence of the Afadin-CC region is highly charged and does not maintain this pattern of hydrophobic residues (*Figure 1F*), suggesting that it may not be a bona fide CC. To test whether the Afadin-CC forms higher oligomers, the GST tag was removed (*Figure 2—figure supplement 1*) and the Afadin-CC alone was analyzed by size-exclusion chromatography coupled to multiangle light scattering (SEC-MALS). This technique allows for the direct measurement of the molar mass of a particle in solution and demonstrated that the Afadin-CC is a monomer, with a molecular mass of ~8 kDa (predicted mass = 8.3 kDa, *Figure 2A*). The chromatography trace also revealed a small amount of truncated Afadin-CC, suggesting that following removal of the GST tag this region may be prone to some degradation. To further characterize the interaction of the PTPRK-ICD with Afadin-CC, isothermal titration calorimetry (ITC) experiments were performed to measure the binding affinity

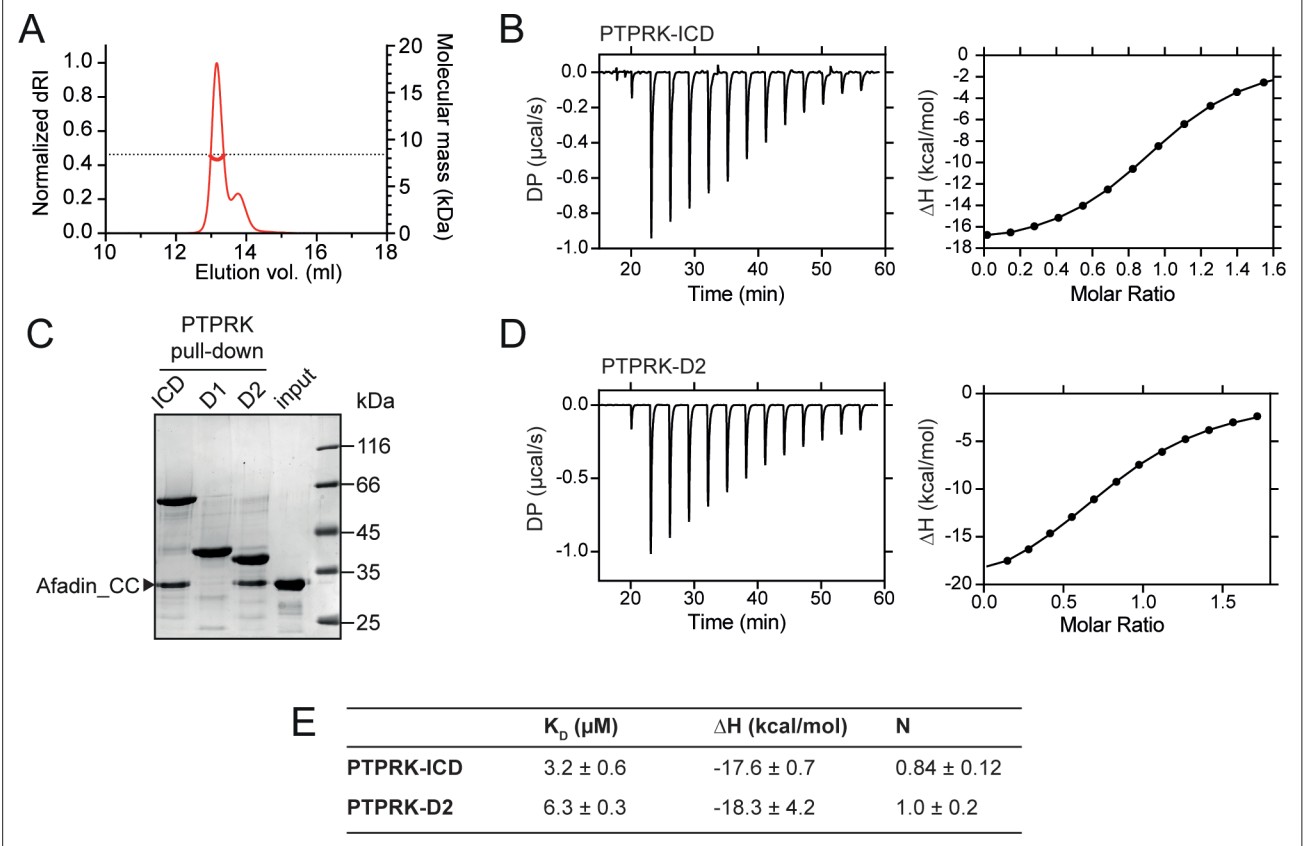

**Figure 2.** PTPRK:Afadin-CC forms an equimolar complex with micromolar affinity. (**A**) Size-exclusion chromatography coupled to multi-angle light scattering (SEC-MALS) analysis of Afadin-CC. The SEC elution profile (normalized differential refractive index [dRI]; thin red line) and weight-averaged molecular mass (red thick line) are shown. The dashed horizontal line indicates the predicted mass of monomeric Afadin-CC after removal of the GST affinity tag. (**B**) Isothermal titration calorimetry (ITC) titration curves of the interaction between Afadin-CC and PTPRK-ICD. Left: baseline-corrected differential power (DP) plotted over time. Right: normalized binding curve showing the integrated change in enthalpy against the molar ratio. (**C**) PTPRK pull-downs using streptavidin bead-conjugated ICD, D1 or D2 domains against GST-Afadin-CC followed by SDS-PAGE and Coomassie staining. Gel is representative of n ≥ 3 independent experiments. (**D**) ITC titration curves of the interaction between Afadin-CC and PTPRK-D2. Data presented as described for (**B**). (**E**) Table showing the dissociation constant ($K_D$), enthalpy (ΔH), and number of binding sites (N) for the ITC experiments performed in this study. Data represents the mean ± SEM of n = 2 independent experiments.

The online version of this article includes the following source data and figure supplement(s) for figure 2:

**Source data 1.** Uncropped, unedited gels for *Figure 2C*.

**Figure supplement 1.** GST-tag removal from Afadin-CC for size-exclusion chromatography coupled to multi-angle light scattering (SEC-MALS) and isothermal titration calorimetry (ITC) experiments.

**Figure supplement 1—source data 1.** Uncropped, unedited gels for *Figure 2—figure supplement 1A*.

and stoichiometry. Afadin-CC (without the GST tag) forms an equimolar complex with PTPRK-ICD with a binding affinity in the low micromolar range ($K_D$ = 3.2 ± 0.6 µM, *Figure 2B and E*). This binding affinity is consistent with a reversible, transient protein–protein interaction as expected for an enzyme-substrate complex. The high ΔH of this interaction (–17.6 ± 0.7 kcal/mol) indicates it is enthalpically driven, consistent with the highly charged nature of the Afadin-CC region.

Previous work highlights a critical role for the D2 pseudophosphatase domain in the specific interaction of Afadin with R2B family receptors (*Fearnley et al., 2019*; *Hay et al., 2020*). To investigate the contribution of the D2 domain to Afadin binding, pull-downs against Afadin-CC were performed using biotinylated PTPRK ICD, D1 and D2 domains. The PTPRK-D2 domain alone was sufficient to bind Afadin-CC to similar levels observed for the full ICD (*Figure 2C*). No interaction of Afadin-CC with the PTPRK-D1 domain was observed, highlighting that the D2 domain is necessary and sufficient for the interaction with this part of Afadin. To confirm this, ITC experiments were performed monitoring the binding of the Afadin-CC to the PTPRK-D2 domain. These data produced very similar results to those

for the full ICD, specifically the Afadin-CC forms an equimolar complex with the PTPRK-D2 domain with a binding affinity in the low micromolar range ($K_D$ = 6.3 ± 0.3 μM, *Figure 2D and E*). These data suggest there is only a very modest (twofold) reduction in binding affinity for the D2 domain when compared with the ICD, consistent with the D2 domain being the primary Afadin binding surface.

## Structure prediction of the PTPRK–Afadin interaction

Having mapped the regions of Afadin and PTPRK required for binding, we sought to understand the structural basis of the interaction, but extensive efforts to crystallize the Afadin-CC:PTPRK-D2 complex were unsuccessful. Therefore, we exploited recent advances in deep learning neural network techniques by using AlphaFold2 (AF2)-Multimer to model the complex based on the truncation mapping data. Initial structural predictions were performed using the PTPRK-D2 domain (aa. 1154–1446) with Afadin-CC (aa. 1393–1455) as inputs. The AF2 model of PTPRK-D2 in complex with the full Afadin-CC fragment used for biochemical characterization highlighted that many residues at both the N- and C-termini of the Afadin-CC may not be contributing to the putative interaction interface (*Figure 3—figure supplement 1*). Therefore, further mapping experiments to refine the binding region of the Afadin-CC were conducted. Both N- and C-terminal truncations of GST-Afadin-CC were expressed, purified, and tested in pull-downs for their ability to bind PTPRK (*Figure 3A*). For N-terminal truncations of Afadin-CC, residues 1393–1400, which are predicted to be unstructured and outside the core helix (*Figure 3—figure supplement 1*), were replaced with an equivalent glycine/serine linker sequence to avoid the possibility of steric hindrance with the N-terminal GST-tag during mapping experiments (*Figure 3A*). In agreement with the initial AF2 model, regions at both the N- and C-termini of Afadin-CC were dispensable for the interaction, while disruption of the core sequence encompassing the charged residues (aa. 1408–1448) resulted in a lack of detectable binding (*Figure 3A*). Interestingly, this core charged region of Afadin is almost completely conserved across vertebrates and diverges in *Drosophila melanogaster*, which lacks a full-length PTPRK ortholog (*Figure 3—figure supplement 2*; *Hatzihristidis et al., 2015*).

Based on these additional mapping data, a new AF2 model was generated using the PTPRK-D2 domain with Afadin-CC minimal region (aa. 1408–1448) as inputs. Of the five models calculated, the top four had the Afadin-CC placed in the same position on the surface of the PTPRK-D2 domain (*Figure 3—figure supplement 3*), strongly suggesting that the CC domain adopts a linear alpha-helical structure that stretches across one face of the D2 domain (*Figure 3B*). The top AF2 model (ipTM + pTM score = 0.58) predicts the overall fold of each domain with high confidence and their relative orientations with moderate to high confidence (*Figure 3C*). To test the validity of this model and further explore the residues involved in Afadin binding to PTPRK, a series of single and double mutations were introduced into this region of the Afadin-CC domain. Mutations of Afadin used in these pull-down assays included residues predicted to be at the interaction interface as well as residues away from the interface (*Figure 3D and E*). Double alanine mutation at Afadin residues W1418-Y1419 or R1432-K1433, which contribute large sidechains to the interaction interface, abolishes binding to PTPRK, while E1429-R1430 to alanine reduces but does not abolish binding. Importantly, additional mutations to residues distal from the interface (R1410, Q1443, T1446) do not interfere with binding to PTPRK. These mutational data strongly support the validity of the predicted structure for how Afadin binds to the PTPRK D2 domain.

## Substrate specificity of PTPRK recruitment and dephosphorylation of Afadin

Our previous work demonstrated that both PTPRK and PTPRU bind Afadin while PTPRM cannot (*Fearnley et al., 2019*; *Hay et al., 2020*). Therefore, residues on the D2 domain that are critical for Afadin binding should be conserved in PTPRK and PTPRU but divergent in PTPRM. Based on a multiple-sequence alignment of PTPRK, PTPRU, and PTPRM (*Figure 4—figure supplement 1*), the positions of unique PTPRM residues were mapped onto the AF2 model of PTPRK-D2 (*Figure 4A*, the predicted position of the D1 domain is illustrated for clarity and orientation). Interestingly, a cluster of four residues mapped to a region of the PTPRK-D2 surface that is predicted to be the binding site for Afadin-CC (*Figure 4A*) and are located adjacent to several Afadin-CC residues that we have shown to be necessary for complex formation (*Figure 3D and E*). This binding site is located on the opposite face relative to the cysteine of the would-be 'active' site in the PTPRK-D2 domain (*Figure 4A*).

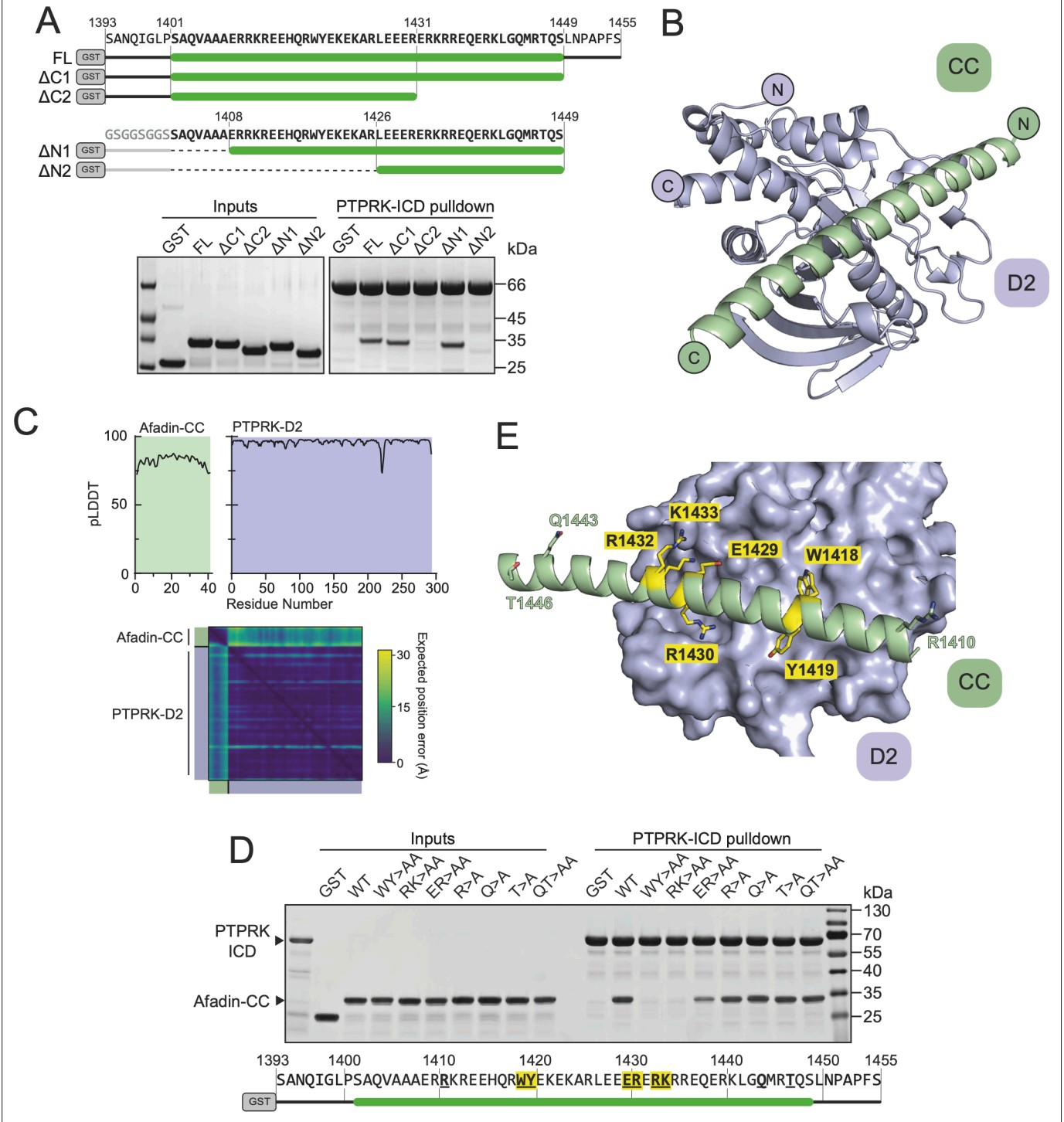

**Figure 3.** Structural prediction of the PTPRK-D2:Afadin-CC complex. (**A**) Top: schematic of different GST-Afadin-CC truncations used for further interaction mapping experiments. Regions within the predicted helix are marked by a green bar. Dashed lines indicate deleted regions in N-terminal truncations. Bottom: pull-downs using streptavidin bead-conjugated PTPRK-ICD against GST-Afadin-CC truncations followed by SDS-PAGE and Coomassie staining. (**B**) The top model generated by AF2-Multimer of the PTPRK-D2 domain (blue) in complex with Afadin-CC (green). (**C**) Prediction quality analysis for the top PTPRK-D2:Afadin-CC complex model. Top: plot of predicted local distance difference test (pLDDT) for Afadin-CC (green) and PTPRK-D2 (blue). Bottom: predicted aligned error (PAE) plot for the PTPRK-D2:Afadin-CC complex. Quality analyses for all five generated models are available in *Figure 3—figure supplement 3*. (**D**) PTPRK pull-downs using streptavidin bead-conjugated PTPRK-ICD against GST-Afadin-CC point mutants followed by SDS-PAGE and Coomassie staining. Residue numbering of the Afadin-CC sequence is shown below. Residues that were mutated are in bold underline, with mutations that alter Afadin-CC binding highlighted in yellow. (**E**) Molecular surface representation of PTPRK-D2 (blue) in

*Figure 3 continued on next page*

*Figure 3 continued*

complex with Afadin-CC (green ribbons). The sidechains of Afadin-CC residues that were mutated in (**D**) are shown in stick representation, with residues that were shown to be critical for PTPRK binding highlighted in yellow. Gels shown in this figure are representative of n ≥ 3 independent experiments.

The online version of this article includes the following source data and figure supplement(s) for figure 3:

**Source data 1.** Uncropped, unedited gels for *Figure 3A*.

**Source data 2.** Uncropped, unedited gel images for *Figure 3D*.

**Figure supplement 1.** Structural prediction of the PTPRK-D2:Afadin-CC complex.

**Figure supplement 2.** Species conservation of Afadin-CC.

**Figure supplement 3.** Prediction quality analysis of AF2 multimer-generated PTPRK-D2:Afadin-CC complex models.

Although these residues of PTPRK (Y1270, G1273, E1332, and L1335) are not all highly charged, within the context of the PTPRK model they form an acidic pocket into which the Afadin-CC is predicted to bind (*Figure 4B and C*). To investigate potential molecular differences at this site, a PTPRM-D2 (aa. 1160–1452) AF2 model was generated to allow comparison to PTPRK-D2. Strikingly, in a model of the PTPRM-D2 the altered residues in this region result in a different surface topology and charge, having lost the acidic pocket observed for PTPRK-D2 (*Figure 4B and C*). These changes in surface properties may therefore account for the differential binding to Afadin.

To further test the validity of the predicted PTPRK-D2:Afadin-CC complex, and to better understand the binding specificity, residues in the PTPRK-D2 acidic pocket were mutated to their PTPRM equivalents. Specifically, PTPRK-G1273 was mutated to histidine (G1273H) and L1335 to arginine (L1335R). These changes to PTPRM-equivalent residues are predicted to cause steric hindrance (G1273H) or introduce a basic residue into the acidic pocket (L1335R). Each of these single-point mutations were sufficient to abolish the interaction of the PTPRK-D2 with the GST-Afadin-CC (*Figure 4D*). Furthermore, the introduction of both mutations (double mutant [DM]) into the PTPRK-D2 or full PTPRK-ICD prevents binding to Afadin-CC (*Figure 4E*). To quantify the loss in affinity due to these point mutations, ITC binding experiments were repeated using the PTPRK-D2-DM with Afadin-CC and no binding was detected, confirming loss of the interaction (*Figure 4F*). To further confirm that these mutations could interfere with Afadin binding in the context of the full-length protein, wheat germ lysate pull-downs were repeated with WT and DM PTPRK constructs. While PTPRK-ICD and PTPRK-D2 bind Afadin, introduction of these two mutations into PTPRK reduces Afadin binding to background levels (*Figure 4G*). Two additional mutations were introduced into the PTPRK-D2 domain to test the predicted complex model. The first, F1225A, alters the PTPRK residue that is predicted to interact with the W1418-Y1419 residues in Afadin that were essential for binding (*Figure 3D*, *Figure 4—figure supplement 2A*). This single-point mutation (F1225A) in PTPRK-D2 vastly reduces binding to the Afadin-CC domain (*Figure 4H*). This residue is conserved across the R2B family so although it contributes to the binding affinity and validates the complex model, it does not contribute to specificity. Another mutation (labeled the 'M-loop') was introduced at the edge of the interface, changing residues CEE in PTPRK (aa. 1372–1374) to the PTPRM-equivalent YNG (aa. 1378–1380) (*Figure 4—figure supplement 2B*). This mutation retains binding, suggesting that this loop does not contribute to specificity of Afadin binding to PTPRK versus PTPRM (*Figure 4H*). The cysteine residue of the M-loop is distal to Afadin in the predicted PTPRK complex and the EE to NG substitution would not introduce significant steric hindrance or charge repulsion, consistent with retained binding. For all mutant forms of the PTPRK-D2 domain, correct folding was confirmed by monitoring their thermal stability and comparing melting temperatures ($T_m$) to wildtype PTPRK-D2 (*Figure 4—figure supplement 3*). Wildtype and mutant PTPRK-D2 domains possessed $T_m$ values between 39 and 41°C, supporting that any loss of binding was not driven by misfolding. These data strongly validate the predicted model for the PTPRK-D2:Afadin-CC complex and identify two key residues within PTPRK D2 that are crucial for binding to full-length Afadin and, based on their lack of conservation in PTPRM, are likely to be important for determining binding specificity.

To determine whether the PTPRK-D2:Afadin-CC interaction is necessary for PTPRK-mediated dephosphorylation of Afadin, dephosphorylation assays were performed using both WT and DM PTPRK-ICDs. Since we and others have observed allosteric modulation of RPTP D1 domains by D2 domains (*Fearnley et al., 2019*; *Wen et al., 2020*), we tested whether introducing D2 domain mutations affects PTPRK-ICD phosphatase activity. The WT and DM PTPRK-ICDs were compared in activity

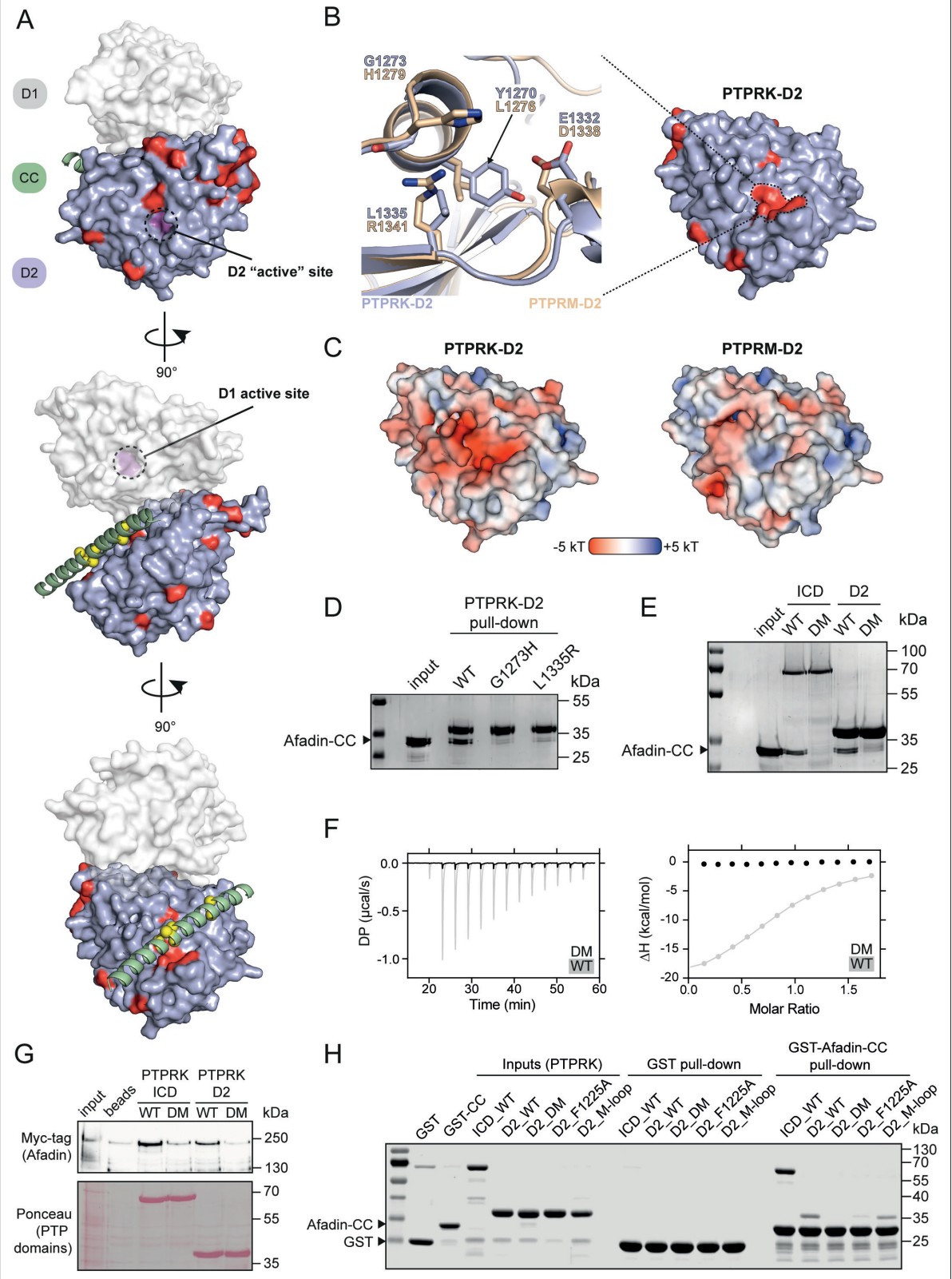

**Figure 4.** PTPRK binds Afadin-CC via an acidic pocket distal from the D2 'active' site. (**A**) Conservation mapping of the PTPRK-D2:Afadin-CC interface. PTPRK-D2 is shown (blue surface representation) in complex with Afadin-CC (green ribbons). Residues that are conserved in both PTPRK/PTPRU but not PTPRM are potentially involved in Afadin binding specificity and are highlighted on the PTPRK-D2 surface in red. Cα atoms of Afadin-CC residues identified as critical for PTPRK binding (see *Figure 3D and E*) are highlighted by yellow spheres. For clarity and orientation, the D1 domain

*Figure 4 continued on next page*

*Figure 4 continued*

of PTPRK has been modeled in transparent surface representation (white) and both D1 and D2 'active' sites highlighted in purple and dotted circles. Three orientations rotated by 90° are shown. (**B**) Inset shows the molecular detail of key residues at the highlighted region on the PTPRK-D2 surface. Equivalent residues for the highlighted region are shown for PTPRK (blue) and PTPRM (wheat) in stick representation. (**C**) Electrostatic properties of PTPRK (left) and PTPRM (right) D2 domains, colored by electrostatic potential (–5 to +5 kT, as red and blue, respectively). Domains are oriented as shown in (**B**). The key PTPRM substitutions highlighted in (**B**) result in altered surface topology and electrostatic potentials compared to PTPRK. (**D**) Pull-downs using streptavidin bead-conjugated WT, G1273H, or L1335R PTPRK-D2 domains against GST-Afadin-CC followed by SDS-PAGE and Coomassie staining. (**E**) Pull-downs using streptavidin bead-conjugated WT and G1273H/L1335R double mutant (DM) PTPRK-ICD and D2 domains against GST-Afadin-CC followed by SDS-PAGE and Coomassie staining. (**F**) Isothermal titration calorimetry (ITC) data showing a lack of interaction between Afadin-CC and PTPRK-D2 DM. Left: baseline-corrected differential power (DP) plotted over time. Right: normalized binding curve showing the integrated change in enthalpy against the molar ratio. To highlight lack of binding, DM data (black) is shown superimposed onto the data for the WT D2 domain (gray), which is also shown in *Figure 2D*. (**G**) Immunoblot analysis of streptavidin bead-conjugated PTPRK pull-downs from wheat germ lysate containing full-length Afadin. Both WT and DM PTPRK-ICD and D2 domains were assayed for their ability to bind full-length Afadin. (**H**) GST pull-downs using GST-Afadin-CC against PTPRK-ICD and D2 WT, DM, F1225A, and M-loop, followed by SDS-PAGE and Coomassie staining. Gels and blots shown in this figure are representative of n ≥ 3 independent experiments.

The online version of this article includes the following source data and figure supplement(s) for figure 4:

**Source data 1.** Uncropped, unedited gel for *Figure 4D*.

**Source data 2.** Uncropped, unedited gel for *Figure 4E*.

**Source data 3.** Uncropped, unedited blot for *Figure 4G*.

**Source data 4.** Uncropped, unedited gel images for *Figure 4H*.

**Figure supplement 1.** Mapping of unique PTPRM residues.

**Figure supplement 2.** Localization of additional PTPRK D2 mutations.

**Figure supplement 3.** Thermal stability of PTPRK-D2 mutants.

**Figure supplement 4.** Prediction quality of models prior to biochemical mapping.

assays using the generic phosphatase substrate 4-nitrophenyl phosphate (pNPP), which confirmed that both proteins have identical activity profiles (*Figure 5A*). Dephosphorylation assays were then performed using PTPRK-ICD WT and DM proteins incubated with quenched, pervanadate-treated lysates. Afadin was dephosphorylated at Y1230 by WT PTPRK-ICD, but not PTPRK-ICD DM (*Figure 5B*, *Figure 5—figure supplement 1*). Comparatively, another PTPRK substrate, p120-Catenin, which has been shown previously to be dephosphorylated efficiently by the PTPRK-D1 domain alone, was dephosphorylated by both proteins – independent of D2 domain mutation status. Results using our Afadin-pY1230 antibody were also confirmed by immunoprecipitation (IP) of pTyr proteins from WT and DM dephosphorylation assays (*Figure 5C*). Afadin was depleted in pTyr-IPs from lysates treated with WT PTPRK-ICD, showing dephosphorylation had occurred, while the levels of phosphorylated Afadin remained unchanged in PTPRK-ICD DM-treated samples (*Figure 5C*). Again, both WT and DM PTPRK were equivalently able to dephosphorylate p120-Catenin. These data show that D2 domain recruitment of Afadin is critical for efficient dephosphorylation by the D1 domain, and that PTPRK uses distinct surfaces to recruit different substrates for dephosphorylation.

## Discussion

Here we demonstrate direct and specific binding of the substrate Afadin to the receptor phosphatase PTPRK. This interaction is mediated via a single helix within an extended disordered region of Afadin that binds directly to the pseudophosphatase D2 domain of PTPRK. Experimental mapping of the binding interface involved truncation mapping of Afadin using different protein production strategies. The cell-free wheat germ-based system allowed for the efficient testing of large Afadin constructs and the rapid mapping down to a region more compatible with large-scale cell-based protein production, such as *E. coli*. This additional mapping allowed for the identification of relevant, smaller regions and domains of both Afadin and PTPRK. By gaining this information experimentally, it was then possible to exploit deep learning techniques via AF2 to make reliable predictions of the interaction interface. Interestingly, if larger regions of either protein, such as the full ICD of PTPRK or longer segments of Afadin, are used for AF2-based complex predictions, a range of different assemblies are predicted with lower confidence (*Figure 4—figure supplement 4*). Therefore, at present, these predictive tools

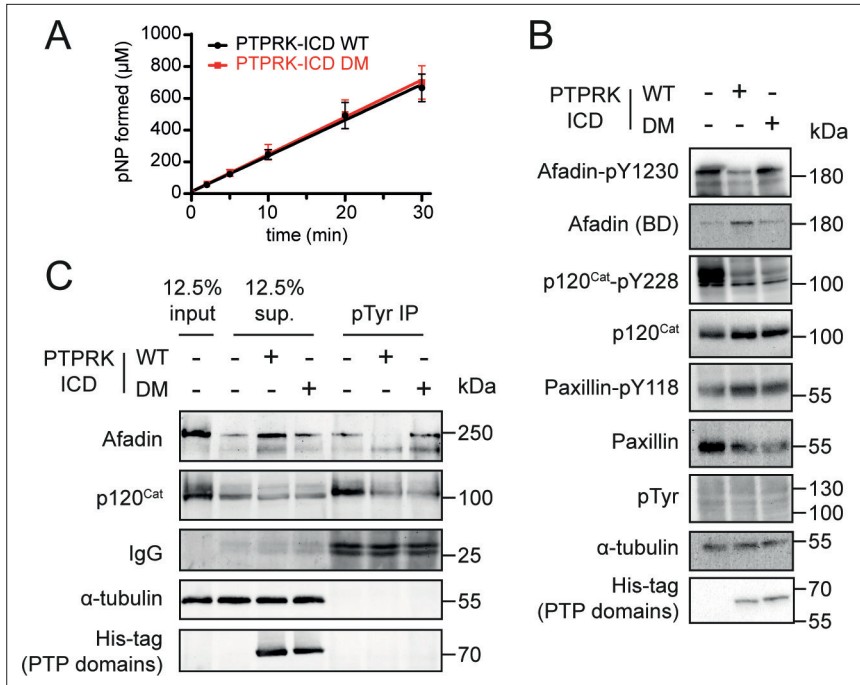

**Figure 5.** Mutation of residues at the interaction interface inhibit dephosphorylation of Afadin by PTPRK. (**A**) Time course of pNPP dephosphorylation by WT (black) and double mutant (DM, red) PTPRK-ICD. Error bars represent ± SEM of n = 2 independent experiments. (**B**) Immunoblot analysis of pervanadate-treated MCF10A lysates incubated for 1.5 hr at 4°C with 0.3 µM of PTPRK-ICD WT or DM. (**C**) Immunoblot analysis of pervanadate-treated MCF10A lysates incubated for 1.5 hr at 4°C with 0.3 µM of PTPRK-ICD WT or DM followed by pTyr immunoprecipitation (IP). Dephosphorylated proteins are depleted from pTyr-IPs and/or enriched in supernatants (sup.). Gels and blots shown in this figure are representative of n ≥ 3 independent experiments.

The online version of this article includes the following source data and figure supplement(s) for figure 5:

**Source data 1.** Uncropped, unedited blots for *Figure 5B*.

**Source data 2.** Uncropped, unedited blots for *Figure 5C*.

**Figure supplement 1.** PTPRK-ICD-DM displays impaired dephosphorylation of Afadin-pY1230, but not p120Cat-pY228.

**Figure supplement 1—source data 1.** Densitometric quantification of *Figure 5—source data 1*, shown in *Figure 5—figure supplement 1*.

are at their most valuable when coupled to experimental approaches that identify the relevant protein fragments to be used as inputs. When this requirement is satisfied, the power of these predictions is clearly demonstrated here by the identification of a testable interface that proved to be highly accurate in informing subsequent experimental assays.

SEC-MALS analysis shows the purified Afadin-CC region that binds PTPRK to be monomeric in solution (*Figure 2A*), demonstrating that binding to PTPRK does not compete with potential dimerization of Afadin via this region. The CC region of Afadin that is shown here to be critical for binding to PTPRK is distal to the tyrosine residue that is targeted for dephosphorylation (Y1230). We also demonstrate that the interaction with the CC region is critical for efficient dephosphorylation, showing that this interaction is important for bringing Afadin within close proximity to PTPRK but does not directly position the relevant pTyr in the active site of the PTPRK-D1 domain. This role for the D2 pseudophosphatase domain in specific substrate recruitment helps to explain the apparent lack of specificity of phosphatases for phosphopeptide substrates, indicating that it is not necessarily a motif near the pTyr that determines substrate targeting.

Our previous observation that Afadin binding is independent of its phosphorylation status (*Fearnley et al., 2019*) is consistent with the data presented here. Specifically, Afadin binds PTPRK via a site that is over 100 amino acids downstream of the target phosphosite and the interaction interface is distinct from the nonfunctional 'active site' on the D2 domain. These data support that the pseudophosphatase

domain has been repurposed in evolution for specific protein–protein interactions. However, it is also possible that even active PTP domains mediate such interactions. Additional substrates, such as PARD3, showed some dependence on the D2 for dephosphorylation by PTPRK and PTPRM, whereas other substrates of PTPRK, such as p120$^{Cat}$, do not require the D2 domain for recruitment (*Fearnley et al., 2019*). This raises interesting questions about simultaneous binding of substrates to PTPRK and potential competition for access to the active site. Furthermore, if the phosphorylation state is not critical for binding, PTPRK might play a scaffolding role connecting multiple junctional components as well as F-actin. Consistent with a scaffolding role for receptor PTPs, structural studies on the R2A receptor PTPRF revealed an interaction interface with Liprin-α spanning both the D1 and D2 domains, adding further complexity to potential binding modes (*Xie et al., 2020*). Interestingly, Liprin is not a PTPRF substrate, thus RPTPs could form multiprotein complexes independent of protein phosphorylation status. This might also help to explain reported phosphatase-independent functions of RPTPs (*Juettner et al., 2019*). Finally, the region of Afadin that interacts with PTPRK has been shown to also mediate binding to αE-catenin (*Maruo et al., 2018*), a core component of the E cadherin-catenin complex (*Niessen and Gottardi, 2008*). Further studies are required to understand how RPTPs regulate cell adhesion while recruiting core adhesion molecules, such as Afadin, potentially in competition with canonical adhesion complexes.

## Methods
### Plasmids and constructs
Amino acid (aa) numbering used throughout is based on the following sequences: PTPRK, UniProt ID: Q15262-3; PTPRM, UniProt ID: P28827-1; Afadin, UniProt ID: P55196-4. All point mutations were generated by site-directed mutagenesis using polymerase chain reaction with Phusion Hot Start II DNA polymerase (Thermo Fisher Scientific). The pET-15b.His.TEV.AviTag expression vectors used for the generation of biotinylated PTPRK-ICD (aa. 864–1439), D1 (aa. 865–1157), and D2 (aa. 1154–1446) domains were generated in a previous study (*Fearnley et al., 2019*). PTPRK-ICD and D2 G1273/L1335R single and double mutants (DM) were generated in these His.TEV.AviTag vectors. For ITC experiments, PTPRK-ICD, D2, and D2-DM were subcloned into a pET-15b expression vector in frame with an N-terminal His6-tag only. For wheat germ cell-free expression, the full-length human Afadin cDNA and truncations (outlined in *Figure 1C*) were subcloned into the pF3A WG (BYDV) vector (Promega) with a modified multiple cloning site (SpeI-AvrII-EcoRI-NheI-SacI-KpnI downstream of an N-terminal Myc-epitope; pF3A-WG-Myc). For bacterial expression, Afadin C-terminal fragments (outlined in *Figure 1D*) were subcloned into the pGEX-6P-1 vector with an in-frame N-terminal GST tag followed by the human rhinovirus 3C protease recognition sequence. For transient overexpression experiments, full-length Afadin was subcloned into pmScarlet-C1 (gift from Z. Kadlecova) and mutated to generate Y1226F and Y1230F single and double (YFYF) mutants.

### Antibodies
All antibodies, except Afadin-pY1230, were used at a 1:1000 dilution in TBS-T (20 mM Tris–HCl, pH 7.4, 137 mM NaCl, 0.2% [v/v] Tween-20) with 3% (w/v) bovine serum albumin (BSA) unless otherwise noted. Rabbit anti-Myc, rabbit anti-Afadin, mouse anti-His-tag, rabbit anti-pTyr, rabbit anti-phospho-p120-Catenin (Y228 and Y904), rabbit anti-paxillin, and rabbit anti-phospho-paxillin (Y118) primary antibodies were purchased from Cell Signaling Technologies. Mouse anti-p120-Catenin and mouse anti-Afadin primary antibodies were purchased from BD Transduction Laboratories. Mouse anti-α-tubulin primary antibody was purchased from Sigma-Aldrich. Mouse anti-RFP antibody was purchased from Thermo Fisher Scientific. Anti-mouse, anti-goat, and anti-rabbit HRP-conjugated secondary antibodies (1:5000 in TBS-T) were purchased from Jackson ImmunoResearch. The rabbit anti-PTPRK antibody was generated previously (*Fearnley et al., 2019*).

The Afadin pTyr-1230 antibody was generated by MRCPPU Reagents and Services, University of Dundee (https://mrcppureagents.dundee.ac.uk/) using a phosphorylated Afadin-Y1230 peptide (CTYTRE[pY]FTFPA) as the antigen. Briefly, the N-terminal Cys residue was used to conjugate the peptide to both keyhole limpet hemocyanin (KLH) and BSA. Sheep were immunized with the antigen followed by four further injections 28 days apart, with bleeds performed 7 days after each injection. Antibodies were affinity purified from serum using the Afadin-Y1230 phospho-peptide. For Western

blotting, the antibody was used at 1 µg/ml in TBS-T with 3% milk, after pre-incubation with 10 µg/ml non-phosphorylated peptide (TYTREYFTFPA) for 30 min at room temperature (RT) or 4°C.

## Protein expression and purification

Expression plasmids were transformed into competent *E. coli* BL21 (DE3) Rosetta cells and cultured in 2× TY medium supplemented with 50 µg/ml ampicillin and 34 µg/ml chloramphenicol at 37°C and 220 RPM to an optical density at 600 nm of 0.6. Cultures were then equilibrated to 20°C and protein expression induced with addition of 1 mM isopropyl-thio-β-D-galactopyranoside for 18 hr. For in vivo biotinylation of Avi-tagged proteins, cultures were supplemented with 200 µM D-biotin (Sigma-Aldrich) at the point of induction. Bacterial cultures were then harvested by centrifugation (4000 × *g*, 20 min) and cell pellets stored at –20°C. Bacterial cell pellets were thawed and resuspended in ice-cold lysis buffer (50 mM Tris–HCl, pH 7.4, 500 mM NaCl, 0.5 mM TCEP, 5% [v/v] glycerol, 1× EDTA-free protease inhibitors) prior to lysis by high-pressure disruption at 25 kpsi (Constant Systems Ltd). Cell lysates were then clarified by centrifugation (40,000 × *g*, 4°C, 30 min).

For GST-Afadin constructs, clarified cell lysates were incubated with washed glutathione sepharose 4B beads (Cytiva) for 1 hr at 4°C. Beads were packed into a gravity column and washed with 20 ml of GST wash buffer (50 mM Tris–HCl, pH 7.4, 500 mM NaCl, 0.5 mM TCEP) and protein eluted in 5 × 1 ml fractions of GST wash buffer supplemented with 50 mM reduced glutathione. Eluted proteins were further purified by SEC or dialysis. SEC was performed using a HiLoad Superdex 200 pg 16/600 column equilibrated in 50 mM Tris–HCl, 150 mM NaCl, 5% [v/v] glycerol, 5 mM DTT. Dialysis was performed using a Slide-A-Lyzer 20K MWCO cassette (Thermo Fisher Scientific) in SEC buffer (50 mM Tris–HCl pH 7.4, 150 mM NaCl, 5% [v/v] glycerol, 5 mM DTT) for 2 hr at RT, followed by overnight at 4°C, and another 2 hr with fresh SEC buffer to remove GSH. The purified protein was recovered from the cassettes and immediately used for streptavidin pull-downs.

For His6-tagged PTP constructs, clarified cell lysates were incubated with washed Ni-NTA agarose beads (QIAGEN) for 1 hr at 4°C. Beads were packed into a gravity column and washed with 10 ml of Ni-NTA wash buffer (50 mM Tris–HCl, pH 7.4, 500 mM NaCl, 5% [v/v] glycerol, 0.5 mM TCEP) with 10 mM imidazole, followed by 10 ml of Ni-NTA wash buffer with 20 mM imidazole. Protein was eluted in 6 × 0.5 ml fractions of elution buffer (50 mM Tris–HCl, pH 7.4, 150 mM NaCl, 5% [v/v] glycerol, 0.5 mM TCEP, 200 mM imidazole). For His6-tag-only constructs used in ITC experiments, proteins were further purified by anion-exchange chromatography (AEX). For PTPRK-D2, eluted Ni-NTA fractions were pooled and then diluted to low-salt/imidazole concentration by 1:10 dilution with dilution buffer (50 mM Tris–HCl, pH 7.4, 5% [v/v] glycerol, 5 mM DTT). The diluted sample was applied to a MonoQ 5/50 GL AEX column (Cytiva), equilibrated in low-salt buffer (50 mM Tris–HCl, pH 7.4, 10 mM NaCl, 5% [v/v] glycerol, 5 mM DTT) and bound protein eluted using a linear gradient with high-salt buffer (50 mM Tris–HCl, pH 7.4, 1 M NaCl, 5% [v/v] glycerol, 5 mM DTT). For PTPRK-ICD, AEX purification was performed as above, but with all buffers adjusted to pH 7.0.

For biotinylated Avi-tag constructs for use in pull-down assays, eluted Ni-NTA fractions were purified by SEC on a HiLoad Superdex 75 (D1 and D2 domains) or 200 (ICDs) pg 16/600 column equilibrated in 50 mM Tris, pH 7.4, 150 mM NaCl, 5% (v/v) glycerol, 5 mM DTT. Routinely, purity of all constructs was assessed by SDS-PAGE followed by staining with Coomassie.

## Assessment of recombinant protein biotinylation

To assess biotinylation, 10 µg of in vivo biotinylated protein was solubilized in an appropriate volume of 5× SDS loading buffer and heated to 95°C for 5 min. Samples were allowed to cool to RT prior to addition of a threefold molar excess of streptavidin (IBA Lifesciences). Samples were incubated for 10 min at RT, then immediately analyzed by SDS-PAGE followed by Coomassie staining. Approximate levels of biotinylation were calculated by 2D densitometry of protein band depletion upon addition of streptavidin using Fiji (*Schindelin et al., 2012*).

## pNPP phosphatase activity assay

Recombinant PTP domains were made up to 500 µl in assay buffer (50 mM Tris–HCl, pH 7.4, 150 mM NaCl, 5% [v/v] glycerol, 5 mM DTT) at 0.6 µM. PTP domains and 20 mM pNPP substrate (in assay buffer) were equilibrated to 30°C for 15 min in an orbital shaking heat block at 500 RPM. To initiate reactions, 500 µl of 20 mM pNPP substrate was added to PTP containing tubes (0.3 µM PTP and

10 mM pNPP final concentrations). Reactions were carried out for 30 min at 30°C in an orbital shaking heat block at 500 RPM. At each time point (0, 2, 5, 10, 20, and 30 min), 100 µl of the total reaction was transferred to a 96-well microplate well containing 50 µl 0.58 M NaOH, terminating the reaction. After the final time point, absorbance of each sample was measured at 405 nm in a Spectramax M5 plate reader (Molecular Devices). Product formation was calculated by interpolation of absorbance values using a 4-nitrophenol standard curve of known concentration.

## Protein pull-downs using wheat germ cell-free expression

Wheat germ reactions were performed using the TnT SP6 High-Yield Wheat Germ Protein Expression System (Promega). On ice, 3 µg of the relevant WG expression plasmid in a total volume of 20 µl ddH$_2$O was combined with 30 µl of freshly thawed wheat germ mastermix. Reactions were carried out for 2 hr at 25°C in a thermocycler.

For wheat germ pull-down assays, 0.5 nmol recombinant in vivo biotinylated PTP domain (36 or 18 µg for ICD and D1/D2 domains, respectively) was bound to 125 µl streptavidin-coated magnetic beads in a total volume of 1 ml of purification buffer (50 mM Tris–HCl, pH 7.4, 150 mM NaCl, 5% [v/v] glycerol) for 1 hr at 4°C with rotation. Beads were collected on a magnetic stand and washed twice with purification buffer, followed by two washes in pull-down buffer (50 mM Tris–HCl, pH 7.4, 150 mM NaCl, 10% [v/v] glycerol, 1% [v/v] Triton X-100). Wheat germ reactions were mixed thoroughly and diluted to a total volume of 200 µl in pull-down buffer. Pull-downs were carried out with 100 µl diluted wheat germ reaction combined with protein-conjugated streptavidin beads in a total volume of 200 µl (approximately 2.5 µM final bait concentration) pull-down buffer in 96-well microplate wells, for 1 hr at RT on a high-speed orbital shaking platform, 700 rpm. Magnetic beads were collected on a 96-well magnetic stand and supernatants removed. To wash, beads were resuspended in 200 µl of pull-down buffer containing 250 mM NaCl followed by shaking for 1 min at 700 rpm. This step was repeated for a total of four washes. After the final wash, samples were transferred to 1.5 ml tubes on a vertical magnetic stand. Beads were rinsed once with 1 ml TBS, then resuspended in 80 µl TBS with 20 µl 5× SDS sample buffer supplemented with 7 mM biotin. Pull-downs were eluted by incubation at 95°C for 10 min, beads separated on a magnetic stand and supernatants collected. Routinely, 50 µl of the final 100 µl sample were analyzed by SDS-PAGE followed by immunoblotting.

## Recombinant protein pull-downs

For PTPRK pull-downs from GST-fusion preys, recombinant in vivo biotinylated PTP domains were bound to streptavidin magnetic beads as described for wheat germ-based pull-downs (see above). Pull-downs were performed using 0.5 nmol PTPRK bait (1 µM) combined with 1 nmol (2 µM) Afadin GST-fusions in a total volume of 500 µl of pull-down buffer (50 mM Tris–HCl, pH 7.4, 150 mM NaCl, 10% [v/v] glycerol, 1% [v/v] Triton X-100), incubated for 1 hr with end-over-end rotation at 4°C. Beads were collected on a magnetic stand and supernatants removed. Pull-downs were washed by thorough resuspension in 500 µl pull-down buffer for a total of four washes. Beads were rinsed once with 1 ml TBS, followed by resuspension in 80 µl TBS with 20 µl 5× SDS sample buffer supplemented with 7 mM biotin. Pull-downs were eluted by incubation at 95°C for 10 min, beads separated on a magnetic stand and supernatants collected. Routinely, 50 µl of the final 100 µl sample were analyzed and visualized by Coomassie staining.

For GST-Afadin pull-downs from PTPRK preys, 1 nmol of GST-Afadin-CC was bound to 30 µl glutathione sepharose beads for 1 hr at 4°C with rotation. Bead-bound GST-Afadin-CC was combined with 0.5 nmol PTPRK proteins in a total volume of 500 µl of pull-down buffer (2 µM GST-Afadin, 1 µM PTPRK final concentration) and pull-downs incubated for 1 hr at 4°C with end-over-end rotation. Beads were collected by brief centrifugation and pull-downs washed by thorough resuspension in 500 µl of pull-down buffer. After a total of four washes, pull-downs were eluted by boiling at 95°C in 1× SDS sample buffer in TBS.

## AlphaFold2-Multimer structure predictions

All AF2 models were generated using default parameters and run via a locally installed version of AF2-Multimer (version 2.1.0; *Evans et al., 2022*; *Jumper et al., 2021*). All models and associated statistics have been deposited in the University of Cambridge Data Repository (https://doi.org/10.17863/CAM.82741). Graphical figures were rendered in PyMOL (Schrödinger LLC).

## Size-exclusion chromatography coupled to multi-angle light scattering (SEC-MALS)

For SEC-MALS analysis, the GST-tag was removed from Afadin-CC by PreScission (3C) protease cleavage, using 30 μg GST-3C per 1 mg GST-Afadin-CC in 50 mM Tris, pH 7.4, 150 mM NaCl, 5% (v/v) glycerol, 5 mM DTT, 0.5 mM EDTA. Cleavage reactions were performed for 16 hr at 4°C, followed by removal of GST-3C and cleaved GST by reabsorption onto glutathione sepharose beads for 1 hr at 4°C. Cleavage reactions were then passed through a gravity column to separate sepharose beads from eluate containing the cleaved Afadin-CC. Afadin-CC was immediately loaded onto a HiLoad Superdex 75 pg 16/600 column (Cytiva) equilibrated in SEC buffer (50 mM Tris–HCl, pH 7.4, 150 mM NaCl, 5% [v/v] glycerol, 5 mM DTT) and peak fractions concentrated in 3K MWCO centrifugal filter unit (Merck Millipore) to 1 mg/ml (120 μM).

A 100 μl sample at 1 mg/ml was injected onto a Superdex 75 Increase 10/300 GL column (Cytiva) equilibrated in SEC buffer at a flow rate of 0.5 ml/min. Data for static light scattering and differential refractive index were measured in-line using DAWN 8+ and Optilab T-rEX detectors, respectively (both Wyatt Technology). The absolute molar masses of the elution peaks were calculated in ASTRA 6 (Wyatt Technology) using a protein dn/dc value of 0.185 ml/g.

## Isothermal titration calorimetry (ITC)

For ITC experiments, the GST-tag was removed from Afadin-CC as described for SEC-MALS experiments (see above), with final SEC purification into ITC buffer (50 mM HEPES, pH 7.4, 150 mM NaCl, 0.5 mM TCEP). PTP domains were exchanged into ITC buffer by SEC using a HiLoad Superdex 200 pg 16/600 (PTPRK-ICD) or Superdex 75 Increase 10/300 (PTPRK-D2) column. ITC experiments were performed on an automated MicroCal PEAQ-ITC (Malvern Panalytical) at 25°C. Syringe titrant (Afadin-CC) at 260 μM was titrated into 30 μM cell titrate (PTPRK-ICD and D2) as 13 × 3 μl injections. ITC data were analyzed and fit to a one-site binding model using the MicroCal PEAQ-ITC analysis software (Malvern Panalytical).

## Differential scanning fluorimetry (DSF)

DSF was performed using Sypro Orange dye (Invitrogen) as per manufacturer's protocol in a Bio-Rad Real-Time PCR system. Reaction mixes consisting of 5 μg recombinant protein and 1× Sypro Orange dye were made up to a total volume of 20 μl in 1× PBS. Samples were then heated on a 1°C per 10 s gradient from 25 to 95°C and protein unfolding at each temperature monitored by measurement of fluorescence using the FRET channel. Fluorescent signal vs. temperature was fitted to a nonlinear Boltzmann-sigmoidal regression in GraphPad Prism, with the $T_m$ calculated from the inflection point of the fitted curve.

## Cells and cell culture

MCF10A cells were obtained from the American Type Culture Collection (ATCC, CRL-10317). HEK293T cells were obtained from D Ron. Cells were maintained in a 37°C humidified 5% $CO_2$ ventilator and passaged every 2–4 days depending on the cell line. MCF10A cells were grown in MCF10A culture medium (1:1 DMEM:Ham's F12, supplemented with 5% [v/v] horse serum, 20 ng/ml EGF, 0.5 mg/ml hydrocortisone, 100 ng/ml cholera toxin, and 10 mg/ml insulin) (*Debnath et al., 2003*). PTPRK-KO MCF10A cells and doxycycline-inducible stable cell lines were generated in a previous study (*Fearnley et al., 2019*) and cultured as for WT cells. HEK293T cells were cultured in DMEM containing 10% (v/v) FBS (Sigma-Aldrich), 2 mM L-glutamine (Sigma-Aldrich). For doxycycline induction of MCF10A cell lines, $2 \times 10^6$ were seeded per 10 cm dish and cultured for 6 days, with media changes on days 2 and 4. Doxycycline (Sigma-Aldrich) was included on day 4, 48 hr prior to lysis. Cell lines that were not obtained from commercial sources were subjected to Mycoplasma testing using MycoAlert PLUS (Lonza) or MycoProbe (R&D Systems) Mycoplasma Detection Kits.

## Lipid-based transfection of Afadin constructs for overexpression

$6 \times 10^5$ HEK293T cells were reverse transfected with mScarlet or mScarlet-Afadin constructs using GeneJuice (Merck Millipore #70967-3). For each well of a 6-well plate, 6 μl GeneJuice transfection reagent was incubated with 200 μl serum/antibiotic-free OptiMEM (Thermo Fisher Scientific) for 5 min

at RT, then combined with 2 µg of plasmid DNA for a further 15 min at RT prior to addition to cells in full growth media. After 4 hr, cells underwent a media change.

## Calf intestinal alkaline phosphatase treatment

HEK293T cells were transiently transfected with indicated mScarlet constructs, pervanadate treated and lysed in RIPA buffer (50 mM Tris–HCl pH 7.5, 150 mM NaCl, 1% [v/v] NP-40, 0.5% [w/v] sodium deoxycholate, 1 mM EDTA, 0.2% [w/v] SDS), cOmplete Protease Inhibitor Cocktail (Roche) and phosphatase inhibitor phosSTOP (Roche). Lysates were cleared by centrifugation at 13,000 × $g$ for 15 min at 4°C. Supernatant protein concentration was determined by BCA assay. 50 µg protein was resuspended with 50 U calf intestinal alkaline phosphatase (CIP) and 1× dephosphorylation buffer (Invivogen #18009-019). Samples were incubated for 40 min at 37°C with gentle agitation. The temperature was increased to 65°C for 10 min to inactivate CIP. 1× sample buffer was added to samples prior to SDS-PAGE and immunoblotting.

## Lipid-based transfection of siRNA pools

$6 \times 10^5$ HEK293T cells were reverse transfected with nontargeting or *AFADIN* siRNA duplexes (ON-TARGETplus SMARTpool, Dharmacon Horizon Discovery) using lipofectamine RNAiMAX as per the manufacturer's instructions (Thermo Fisher Scientific # 13778030). Briefly, for each well of a 6-well plate, 6 µl RNAiMAX was used to transfect 10 nM siRNA duplexes. After 24 hr, media was replaced with complete growth medium and left to recover for another 24 hr prior to cell treatments and processing for analysis.

## Generation of pervanadate-treated lysates

Pervanadate was generated based on *Huyer et al., 1997*. To avoid decomposition, it was prepared immediately prior to use. In a 1.5 ml tube, 100 µl of 100 mM sodium orthovanadate (Alfa Aesar) was combined with 103 µl 0.49 M $H_2O_2$ in 20 mM HEPES, pH 7.3, mixed by gentle inversion and incubated at RT for 5 min. Excess $H_2O_2$ was quenched by addition of 23 µl of 0.5 mg/ml catalase in 50 mM potassium phosphate, followed by mixing by gentle inversion. This yields a 44.2 mM pervanadate stock solution.

To generate pervanadate-treated lysates, MCF10A cells were seeded at $2 \times 10^6$ in a 10 cm dish and cultured until confluent. Culture medium was then aspirated and cells treated with 8 ml of fresh MCF10A growth medium (see 'Cells and cell culture') supplemented with 100 µM pervanadate. Cells were treated for 30 min in a 37°C, 5% $CO_2$ incubator, then moved to ice and washed twice with ice-cold PBS. Each dish was lysed in 600 µl lysis buffer (50 mM Tris–HCl, pH 7.4, 150 mM NaCl, 10% [v/v] glycerol, 1% [v/v] Triton X-100, 1 mM EDTA, 5 mM iodoacetamide [IAA], 1 mM sodium orthovanadate, 10 mM NaF, 1× EDTA-free protease inhibitor) on ice, with periodic agitation, in the dark at 4°C for 30 min. Lysates were scraped, collected into 1.5 ml tubes, and treated with 10 mM DTT on ice for 15 min. Lysates were cleared by centrifugation (14,000 × $g$, 15 min, 4°C). Supernatants were then collected, snap frozen in liquid nitrogen, and stored at –80°C until use.

## SDS-PAGE and immunoblotting

For Coomassie-stained gels, proteins were resolved on 4–12% Bis-Tris NuPAGE, and for Western blotting samples were run on 8 or 10% SDS-PAGE. Protein was transferred to 0.2 µm nitrocellulose membranes by wet transfer at 70 V for 2 hr at 4°C. Membranes were blocked in 5% (w/v) skimmed-milk in TBS-T (20 mM Tris–HCl, pH 7.4, 137 mM NaCl, 0.2% [v/v] Tween-20), prior to incubation with primary antibody overnight at 4°C followed by incubation with HRP-conjugated secondary antibody for 1 hr at RT. Blots were developed using EZ-ECL solution (Geneflow) and imaged using a Bio-Rad ChemiDoc MP imaging system.

## In lysate dephosphorylation assays

Recombinant PTP domains (300 nM final concentration) were added to 200 µg of pervanadate-treated MCF10A lysate in a total volume of 400 µl wash buffer (50 mM Tris–HCl, pH 7.4, 150 mM NaCl, 10% [v/v] glycerol, 1% [v/v] Triton X-100), supplemented with or without 5 mM DTT. Reactions were incubated with end-over-end rotation for indicated times (0.75–1.5 hr) at 4°C, then either directly analyzed by immunoblotting or subjected to pTyr immunoprecipitation.

For pTyr IP experiments, 400 μl dephosphorylation assays were made up to 0.4% (w/v) SDS, vortexed, and incubated on ice for 5 min to terminate reactions. Reactions were then made up to a total volume of 800 μl (0.2% SDS) in wash buffer and 6 μl of rabbit anti-pTyr antibody added to each sample. Samples were then incubated with end-over-end rotation at 4°C for 3 hr, followed by addition of 40 μl washed protein A agarose. IPs were carried out overnight (>16 hr) with end-over-end rotation at 4°C. Agarose beads were collected by centrifugation (15,000 × g, 30 s) and washed by resuspension in wash buffer. This wash step was repeated for a total of four washes. To elute IPs, beads were resuspended in 80 μl of 2.5× SDS loading buffer (diluted in wash buffer) and boiled for 10 min at 95°C. Supernatants were then collected and immediately analyzed by SDS-PAGE followed by immunoblot.

### Protein multiple-sequence alignment (MSA)

MSAs were generated using Clustal Omega (*Sievers et al., 2011*) and edited using Jalview (*Waterhouse et al., 2009*).

## Acknowledgements

We thank Benjamin Butt for assistance with installing AlphaFold and Leah Wynn for assistance with Afadin mutagenesis. This study was supported by a Sir Henry Dale Fellowship jointly funded by the Wellcome Trust and the Royal Society awarded to HJS (109407/Z/15/Z), which also supports KEM. JED was supported by a Royal Society University Research Fellowship (UF100371) and a Wellcome Trust Senior Research Fellowship (219447/Z/19/Z). IMH was funded by a CIMR PhD studentship and Babraham Institutional bridging funds. TL was funded by a PhD jointly funded by Trinity College, Cambridge, and the BBSRC Cambridge SBS-DTP. HJS is an EMBO Young Investigator and Lister Institute Prize Fellow. A Titan V graphics card used for this research was donated to SCG by the NVIDIA Corporation. For the purpose of open access, the author has applied a Creative Commons Attribution (CC BY) license to any Author Accepted Manuscript version arising from this submission.

## Additional information

### Funding

| Funder | Grant reference number | Author |
| --- | --- | --- |
| Wellcome and Royal Society | 109407/Z/15/Z | Hayley J Sharpe |
| Wellcome Trust | 219447/Z/19/Z | Janet E Deane |
| Royal Society | UF100371 | Janet E Deane |

The funders had no role in study design, data collection and interpretation, or the decision to submit the work for publication. For the purpose of Open Access, the authors have applied a CC BY public copyright license to any Author Accepted Manuscript version arising from this submission.

### Author contributions

Iain M Hay, Data curation, Formal analysis, Investigation, Visualization, Methodology, Writing - original draft, Writing – review and editing; Katie E Mulholland, Investigation, Writing – review and editing; Tiffany Lai, Formal analysis, Investigation, Methodology, Writing – review and editing; Stephen C Graham, Software, Methodology, Writing – review and editing; Hayley J Sharpe, Conceptualization, Supervision, Funding acquisition, Methodology, Writing - original draft, Writing – review and editing; Janet E Deane, Conceptualization, Supervision, Methodology, Writing - original draft, Writing – review and editing

### Author ORCIDs

Iain M Hay http://orcid.org/0000-0002-5451-1768
Tiffany Lai http://orcid.org/0000-0003-2451-0892
Stephen C Graham http://orcid.org/0000-0003-4547-4034
Hayley J Sharpe http://orcid.org/0000-0002-4723-298X

Janet E Deane http://orcid.org/0000-0002-4863-0330

**Decision letter and Author response**
Decision letter https://doi.org/10.7554/eLife.79855.sa1
Author response https://doi.org/10.7554/eLife.79855.sa2

## Additional files

### Supplementary files
• Transparent reporting form

### Data availability
All structural models predicted using AlphaFold have been deposited in the University of Cambridge Data Repository: https://doi.org/10.17863/CAM.82741.

The following dataset was generated:

| Author(s) | Year | Dataset title | Dataset URL | Database and Identifier |
|---|---|---|---|---|
| Deane J, Hay I, Graham S, Sharpe H | 2022 | AlphaFold2 Multimer models of PTPRK and Afadin domains | https://doi.org/10.17863/CAM.82741 | Apollo - University of Cambridge Repository, 10.17863/CAM.82741 |

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

# Appendix 1

### Appendix 1—key resources table

| Reagent type (species) or resource | Designation | Source or reference | Identifiers | Additional information |
|---|---|---|---|---|
| Gene (human) | *PTPRK* | | ENSEMBL: ENSG00000152894 | |
| Gene (human) | *AFDN* | | ENSEMBL: ENSG00000130396 | |
| Cell line (human) | MCF10A | ATCC | CRL-10317 | |
| Cell line (human) | HEK293T | D. Ron | N/A | |
| Transfected construct (human) | MCF10A PTPRK KO pooled.tGFP | *Fearnley et al., 2019* | N/A | Lentivirally transduced stable cell line |
| Transfected construct (human) | MCF10A PTPRK KO pooled.tGFP. P2A.PTPRK | *Fearnley et al., 2019* | N/A | Lentivirally transduced stable cell line |
| Transfected construct (human) | MCF10A PTPRK KO pooled.tGFP.P2A. PTPRK.C1089S | *Fearnley et al., 2019* | N/A | Lentivirally transduced stable cell line |
| Transfected construct (human) | MCF10A tGFP | *Fearnley et al., 2019* | N/A | Lentivirally transduced stable cell line |
| Antibody | Anti-PTPRK (rabbit monoclonal) | *Fearnley et al., 2019* | 2.H4 | Western blot: 1:1000 |
| Antibody | Anti-Afadin-pY1230 (sheep polyclonal) | This study | N/A | Characterized in *Figure 1—figure supplement 2* Western blot: 1 µg/ml (with 10 µg/ml non-phosphopeptide) Available on request from Sharpe lab, Babraham Institute |
| Antibody | Anti-Afadin (mouse monoclonal) | BD Transduction Laboratories | Cat#610732 | Western blot: 1:1000 |
| Antibody | Anti-p120 catenin (mouse monoclonal) | BD Transduction Laboratories | Cat#610133 | Western blot: 1:1000 |
| Antibody | Anti-RFP (mouse monoclonal) | Thermo Fisher Scientific | Cat#MA5-15257 | Western blot: 1:1000 |
| Antibody | Anti-Turbo-GFP (mouse monoclonal) | OriGene | TA150041 | Western blot: 1:1000 |
| Antibody | Anti-Afadin (rabbit monoclonal) | Cell Signaling Technology | Cat#13531 | Western blot: 1:1000 |
| Antibody | Anti-His (mouse monoclonal) | Cell Signaling Technology | Cat#2366 | Western blot: 1:1000 |
| Antibody | Anti-phospho-tyrosine (P-Tyr-1000) (rabbit monoclonal) | Cell Signaling Technology | Cat#8954 | Western blot: 1:2000 |
| Antibody | Anti-paxillin (rabbit monoclonal) | Cell Signaling Technology | Cat#12065 (D9G12) | Western blot: 1:1000 |
| Antibody | Anti-phospho-p120 catenin (Y904) (rabbit polyclonal) | Cell Signaling Technology | Cat#2910 | Western blot: 1:1000 |
| Antibody | Anti-phospho-p120 catenin (Y228) (rabbit polyclonal) | Cell Signaling Technology | Cat#2911 | Western blot: 1:1000 |
| Antibody | Anti-phospho-paxillin (Y118) | Cell Signaling Technology | Cat#2541 | Western blot: 1:1000 |

*Appendix 1 Continued on next page*

*Appendix 1 Continued*

| Reagent type (species) or resource | Designation | Source or reference | Identifiers | Additional information |
|---|---|---|---|---|
| Antibody | Anti-Tubulin (alpha) (mouse monoclonal) | Sigma | Cat#T6199 | Western blot: 1:1000 |
| Antibody | HRP-conjugated-donkey anti-goat IgG | Jackson ImmunoResearch | Cat#705-035-147 | Western blot: 1:5000 |
| Antibody | HRP-conjugated-donkey anti-rabbit IgG | Jackson ImmunoResearch | Cat#711-035-152 | Western blot: 1:5000 |
| Antibody | HRP-conjugated-donkey anti-mouse IgG | Jackson ImmunoResearch | Cat#711-035-152 | Western blot: 1:5000 |
| Antibody | HRP-conjugated-mouse anti-rabbit IgG (conformation specific) | Cell Signaling Technology | Cat#5127S | Western blot: 1:2000 |
| Recombinant DNA reagent | pET15b | J. Deane | N/A | |
| Recombinant DNA reagent | pET15b.His.TEV.Avi | *Fearnley et al., 2019* | N/A | |
| Recombinant DNA reagent | pET15b.His.TEV.Avi.PTPRK.ICD | *Fearnley et al., 2019* | UniProt: Q15262-3 | |
| Recombinant DNA reagent | pET15b.His.TEV.Avi.PTPRK.ICD.D1057A | *Fearnley et al., 2019* | UniProt: Q15262-3 | |
| Recombinant DNA reagent | pET15b.His.TEV.Avi.PTPRK.ICD.C1089S | *Fearnley et al., 2019* | UniProt: Q15262-3 | |
| Recombinant DNA reagent | pET15b.His.TEV.Avi.PTPRK.D1 | *Fearnley et al., 2019* | UniProt: Q15262-3 | |
| Recombinant DNA reagent | pET15b.His.TEV.Avi.PTPRK.D2 | *Fearnley et al., 2019* | UniProt: Q15262-3 | |
| Recombinant DNA reagent | PET15b.His.TEV.Avi.PTPRK.D2.G1273H | This study | UniProt: Q15262-3 | Mutations: G1273H<br><br>See *Figure 4* |
| Recombinant DNA reagent | pET15b.His.TEV.Avi.PTPRK.D2.L1335R | This study | UniProt: Q15262-3 | Mutations: L1335R<br><br>See *Figure 4D* |
| Recombinant DNA reagent | pET15b.His.TEV.Avi.PTPRK.ICD.DM | This study | UniProt: Q15262-3 | Mutations: G1273H L1335R<br>See *Figure 4E* |
| Recombinant DNA reagent | pET15b.His.TEV.Avi.PTPRK.D2.DM | This study | UniProt: Q15262-3 | Mutations: G1273H L1335R<br>See *Figure 4E* |
| Recombinant DNA reagent | pET15b.His.TEV.Avi.PTPRK.D2.F1225A | This study | UniProt: Q15262-3 | Mutation: F1225A<br>See *Figure 4H* |
| Recombinant DNA reagent | pET15b.His.TEV.Avi.PTPRK.D2.M-loop | This study | UniProt: Q15262-3 | Mutations: C1372Y E1373N E1374G<br>See *Figure 4H* |
| Recombinant DNA reagent | pGEX-6P-1 | J. Deane | N/A | |
| Recombinant DNA reagent | pGEX-6P-Afadin-1098-C* | This study | UniProt: P55196-4 | See *Figure 1E* |
| Recombinant DNA reagent | pGEX-6P-Afadin-1098-1507 | This study | UniProt: P55196-4 | See *Figure 1E* |
| Recombinant DNA reagent | pGEX-6P-Afadin-1098-1407 | This study | UniProt: P55196-4 | See *Figure 1* |
| Recombinant DNA reagent | pGEX-6P-Afadin-1514-C* | This study | UniProt: P55196-4 | See *Figure 1E* |

*Appendix 1 Continued*

| Reagent type (species) or resource | Designation | Source or reference | Identifiers | Additional information |
|---|---|---|---|---|
| Recombinant DNA reagent | pGEX-6P-Afadin-1393-C* | This study | UniProt: P55196-4 | See *Figure 1E* |
| Recombinant DNA reagent | pGEX-6P-Afadin-1393-1507 | This study | UniProt: P55196-4 | See *Figure 1E* |
| Recombinant DNA reagent | pGEX-6P-Afadin-CC | This study | UniProt: P55196-4 | Encoding amino acids: 1393–1455<br><br>See *Figure 1F* |
| Recombinant DNA reagent | pGEX-6P-Afadin-CC-WY>AA | This study | UniProt: P55196-4 | Mutations: W1418A Y1419A<br>See *Figure 3D* |
| Recombinant DNA reagent | pGEX-6P-Afadin-CC-ER>AA | This study | UniProt: P55196-4 | Mutations: E1429A R1430A<br>See *Figure 3D* |
| Recombinant DNA reagent | pGEX-6P-Afadin-CC-RK>AA | This study | UniProt: P55196-4 | Mutations: R1432A K1433A<br>See *Figure 3D* |
| Recombinant DNA reagent | pGEX-6P-Afadin-CC-Q>A | This study | UniProt: P55196-4 | Mutations: Q1443A<br>See *Figure 3D* |
| Recombinant DNA reagent | pGEX-6P-Afadin-CC-T>A | This study | UniProt: P55196-4 | Mutations: T1446A<br>See *Figure 3D* |
| Recombinant DNA reagent | pGEX-6P-Afadin-CC-QT>AA | This study | UniProt: P55196-4 | Mutations: Q1443A T1446A<br>See *Figure 3D* |
| Recombinant DNA reagent | pmScarlet-C1 | Z. Kadlecova | N/A | |
| Recombinant DNA reagent | pmScarlet-Afadin | This study | UniProt: P55196-4 | See *Figure 1—figure supplement 2D* |
| Recombinant DNA reagent | pmScarlet-Afadin-Y1226F | This study | UniProt: P55196-4 | Mutation: Y1226F<br>See *Figure 1—figure supplement 2D* |
| Recombinant DNA reagent | pmScarlet-Afadin-Y1230F | This study | UniProt: P55196-4 | Mutation: Y1230F<br>See *Figure 1—figure supplement 2D* |
| Recombinant DNA reagent | pmScarlet-Afadin-YF YF | This study | UniProt: P55196-4 | Mutations: Y1226F Y1230F<br>See *Figure 1—figure supplement 2D* |
| Sequence-based reagent | ON-TARGETplus Human AFDN siRNA: | Dharmacon, GE Healthcare | L-020075-02-0005 | |
| Sequence-based reagent | ON-TARGETplus Non-targeting pool siRNA: | Dharmacon, GE Healthcare | Cat#D-001810-10-05 | |
| Peptide, recombinant protein | Catalase | Sigma | Cat#C134514 | |
| Peptide, recombinant protein | Cholera toxin | Sigma | Cat#C-8052 | |
| Peptide, recombinant protein | Insulin | Sigma | Cat#I-1882 | |
| Peptide, recombinant protein | Epidermal growth factor | PeproTech | Cat#AF-100-15-1MG | |
| Commercial assay or kit | Q5 High-Fidelity DNA Polymerase | New England Biolabs | Cat#M0491S | |
| Commercial assay or kit | Phusion Hot Start II DNA polymerase | Thermo Fisher Scientific | Cat#F549L | |
| Commercial assay or kit | EZ-ECL substrate | Geneflow | Cat#K1-0170 | |
| Commercial assay or kit | InstantBlue | Expedeon | Cat#ISB1L | |

*Appendix 1 Continued on next page*

*Appendix 1 Continued*

| Reagent type (species) or resource | Designation | Source or reference | Identifiers | Additional information |
|---|---|---|---|---|
| Commercial assay or kit | Phosphatase inhibitor cocktail | Roche | Cat#04906845001 | |
| Commercial assay or kit | TaqMan Universal Master Mix II | Applied Biosystems | Cat#4440040 | |
| Chemical compound, drug | Hydrogen peroxide | Thermo Fisher Scientific | Cat#H/1750/15 | |
| Chemical compound, drug | Sodium orthovanadate | Alfa Aesar | Cat#J60191 | |
| Chemical compound, drug | 250 kDa-FITC-dextran | Sigma | Cat#FD250S-100MG | |
| Chemical compound, drug | Para-Nitrophenol-phosphate (pNPP) | New England Biolabs | Cat#P0757 | |
| Chemical compound, drug | IPTG | Generon | Cat#GEN-S-02122 | |
| Chemical compound, drug | D-biotin | Sigma | Cat#B4639 | |
| Chemical compound, drug | L-Glutamine | Sigma | Cat#G7513 | |
| Chemical compound, drug | Hydrocortisone | Sigma | Cat#H-0888 | |
| Chemical compound, drug | $NH_4OH$ | Acros Organics | Cat#460801000 | |
| Chemical compound, drug | Methanol-free 16% (w/v) paraformaldehyde (PFA) | Thermo Fisher Scientific | Cat#28906 | |
| Software, algorithm | FIJI/ImageJ | Laboratory for Optical and Computational Instrumentation | | University of Wisconsin-Madison |
| Software, algorithm | GraphPad | Prism | | |
| Software, algorithm | Chimera | UCSF | | |
| Strain, strain background (*Escherichia coli*) | STABLE competent *E. coli* | NEB | Cat#C3040I | |
| Strain, strain background (*E. coli*) | DH5alpha competent *E. coli* | Invitrogen | Cat#18265017 | |
| Strain, strain background (*E. coli*) | BL21 DE3 Rosetta *E. coli* | J. Deane | N/A | |
| Other | DMEM | Thermo Fisher Scientific | Cat#41965-039 | Component of cell culture media |
| Other | Ham's F-12 | Sigma | Cat#N4888 | Component of cell culture media |
| Other | Horse serum | Thermo Fisher Scientific | Cat#16050-122 | Component of cell culture media |
| Other | HRP-conjugated streptavidin | Thermo Fisher Scientific | Cat#434323 | For detection of biotinylated proteins |
| Other | Fetal bovine serum | Sigma | Cat#F7524-500ml | Component of cell culture media |
| Other | Trypsin-EDTA solution | Sigma | Cat#T3924 | Reagent used to lift cells from culture vessel |
| Other | GeneJuice | Merck Millipore | Cat#70967-3 | Transfection reagent |

*Appendix 1 Continued on next page*

*Appendix 1 Continued*

| Reagent type (species) or resource | Designation | Source or reference | Identifiers | Additional information |
|---|---|---|---|---|
| Other | EDTA-free protease inhibitors | Roche | Cat#11836170001 | Component of cell lysis buffer |
| Other | Lipofectamine RNAiMAX | Invitrogen | Cat#13778075 | Transfection reagent |
| Other | OptiMEM | Thermo Fisher Scientific | Cat#31985070 | Component of cell culture media |
| Other | Protein G agarose beads | Merck Millipore | Cat#16-266 | Affinity reagent for immunoprecipitations |
| Other | Ni-NTA agarose | QIAGEN | Cat#1018244 | Affinity reagent for His-tag purification |
| Other | Streptavidin-coated magnetic beads | New England Biolabs | Cat#S1420S | Affinity reagent for Avi-tag pull-downs |
| Other | Streptavidin agarose | Thermo Fisher Scientific | Cat#20357 | Affinity reagent for Avi-tag pull-downs |
| Other | Superdex 200 16/600 column | GE Healthcare | Cat#28-9893-35 | Chromatography column |
| Other | Superdex 75 16/600 column | GE Healthcare | Cat#28-9893-33 | Chromatography column |
| Other | Ultracel-3K regenerated cellulose centrifugal filter | Merck Millipore | Cat#UFC900324 | Chromatography column |
| Other | Ultracel-10K regenerated cellulose centrifugal filter | Merck Millipore | Cat#UFC901024 | Used for protein concentration |
| Other | Ultracel-30K regenerated cellulose centrifugal filter | Merck Millipore | Cat#UFC903024 | Used for protein concentration |
| Other | NuPAGE 4–12% Bis-Tris gel | Thermo Fisher Scientific | Cat#NP0321BOX | SDS PAGE electrophoresis gel |
| Other | Slide-A-Lyzer 20K MWCO | Thermo Fisher Scientific | Cat#66003 | Dialysis cassette |
| Other | SYPRO Orange dye | Thermo Fisher Scientific | Cat#S6650 | Used for thermal shift assays |
| Other | MycoAlert PLUS Mycoplasma Detection Kit | Lonza | #LT07-705 | Used for testing cell lines |
| Other | MycoProbe Mycoplasma Detection Kit | R&D Systems | #CUL001B | Used for testing cell lines |

