## [Editor Report]

This Research Advance follows up on the authors' recent *eLife* article where they reported the identification of a series of pTyr protein targets for the protein–tyrosine phosphatase (PTP) activity of the receptor-like PTP, PTPRK. Here they identify a docking site in the catalytically inactive D2 pseudophosphatase domain that promotes substrate dephosphorylation by the D1 phosphatase domain. The evidence for the docking interaction is convincing, and the study is important because it suggests that the D2 pseudophosphatase domains of other RPTPs may similarly function in substrate recruitment and selectivity. The article will be of interest to biochemists studying protein phosphorylation–dephosphorylation and signal transduction mechanisms.

---

## [Decision Letter]

**Decision letter after peer review:**

Thank you for submitting your article "Molecular mechanism of Afadin substrate recruitment to the receptor phosphatase PTPRK via its pseudophosphatase domain" for consideration by *eLife*. Your article has been reviewed by 3 peer reviewers, including Tony Hunter as Reviewing Editor and Reviewer #1, and the evaluation has been overseen by Jonathan Cooper as the Senior Editor. The following individual involved in review of your submission has agreed to reveal their identity: Stefan Knapp (Reviewer #3).

The reviewers agree that your new insights into how Afadin interacts with the D2 pseudophosphatase domain of PTPRK and how this may serve as a recruitment mechanism through which the D1 active phosphatase domain of PTPRK can dephosphorylate Afadin are of significant interest, particularly because they could be the harbinger of a general mechanism for twin domain RPTP substrate recruitment. However, in the absence of a genuine Afadin CC/PTPRK D2 structure, some additional validation of the proposed Afadin CC/PTPRK D2 interface is required to bolster your conclusions before publication.

The essential revisions requested are:

1. The use of additional mutations of key residues in the predicted Afadin CC/PTPRK D2 interface to functionally validate the structural significance of the proposed Afadin recruitment interaction.

2. Further characterization of the pY1230 antibodies to demonstrate their specificity for pY1230 Afadin.

Some additional suggestions for improvement can be found in the individual reviews, where one particular concern is that the anti-pY1230 Afadin antibodies have not been sufficiently well characterized. We look forward to receiving a revised version of your paper describing this exciting new mechanism in which the poorly understood pseudophosphatase domain in PTPRK acts to recruit pTyr substrate proteins.

*Reviewer #1 (Recommendations for the authors):*

1. The AF2 model of the Afadin 1393-1455 region bound to the PTPRK D2 domain is supported by the properties of the acidic pocket DM D2 mutant, but, in the absence of a co-crystal structure, it would be reassuring if some additional mutations in the D2 domain as well as in the Afadin core binding motif were analyzed.

2. The DM mutant data show that the D2 domain interaction is important for Afadin pTyr dephosphorylation in vitro, but one would also like evidence that the DM PTPRK mutant lacks Afadin pY1230 dephosphorylating activity in cells. The authors have the PTPRK KO MCF10A cells they generated in their first *eLife* paper that could be used to re-express the WT and DM PTPRK and then monitor Afadin dephosphorylation with their new anti-pY1230 antibodies.

3. From the model structure in Figure 3B, it is not clear whether the Afadin helix binds to the backside or frontside of the D2 domain. In other words, where is the Afadin binding pocket relative to the "active" site of D2, and would oxidation of the active cite Cys be expected to alter Afadin binding? Some further discussion is called for.

4. The blotting data for the Afadin band in Figures 5F and G are not very clean and need quantifying. In the experiment in panel F it is not clear why the authors did not immunoblot an Afadin IP with their new pY1230 antibodies, instead of precipitating with anti-pTyr antibodies and then blotting with anti-Afadin antibodies. This would enable them to demonstrate that the Afadin/D2 interaction specifically promotes dephosphorylation of pY1230 as opposed to dephosphorylation of other pTyr residues in Afadin (admittedly panel G suggests that pY1230 is not dephosphorylated by the DM mutant). With regard to these experiments, the authors should indicate in the text that the pervanadate in the lysate was neutralized by addition of an excess of DTT (presumably before addition of the recombinant PTPRK proteins).

5. The anti-pY1230 Afadin antibodies should be more thoroughly characterized for their specificity. For instance, based on the top panel of Figure S2, it is not clear what band the reader is supposed to focus on – presumably the upper band but the "pY1230" signal in this band did not go down as much as the Afadin protein band in the second panel in the Afadin siRNA sample. The authors need to use a Y1230F mutant Afadin and YopH PTP treatment of the pV lysate as a negative controls, etc., to validate the specificity of these antibodies. If the new pY1230 antibodies can be shown to be specific, it would be interesting to know whether they can be used to detect pY1230 phosphorylated Afadin by IF, and whether it colocalizes with PTPRK.

*Reviewer #2 (Recommendations for the authors):*

1. Would recommend reporting in the figure legend how many experimental replicates have been collected for each experiment.

2. Related to #1, it is noted that the mean and SEM in Figure 2E are based on only two replicates.

3. The statement that the binding of the anti-afadin antibody is affected by phosphorylation could be better supported by experiments via transfection of tagged afadin followed by blotting with anti-tag vs anti-afadin antibodies.

4. Is the "reverse" mutation of PTPRM residues homolog to PTPRK aa 1273 and 1335 respectively into H and L sufficient to rescue the interaction with afadin and its dephosphorylation by PTPRM?

5. Could the authors check the numbering of residues in Supplementary Figure 9: residue 1273 appears to be an L in that figure.

*Reviewer #3 (Recommendations for the authors):*

I think this is an interesting study that reports an exciting new mechanism involving the poorly understood pseudophosphatase domain in this PTP. I think however since the foundation of the study heavily relies on in silico predictions, more experimental data should be provided to confirm the docking and interaction hypothesis. Some of the suggested experiments, for instance demonstrating that the CC domain in Afadin is indeed helical would be easy to confirm experimentally. Also, key residues in the substrate should be mutated. In general, including a mutant (Y/P mutant) in Afadin would help to confirm many of the experimental data as pY specific peptide antibodies are notoriously unselective.

---

## [Author Response]

The essential revisions requested are:1. The use of additional mutations of key residues in the predicted Afadin CC/PTPRK D2 interface to functionally validate the structural significance of the proposed Afadin recruitment interaction.

We have now made and tested several additional mutations within both the Afadin-CC and PTPRK-D2 domains to further validate the AF2 predicted model of the complex.

For Afadin-CC we introduced several single and double mutations along the helix including residues predicted to be in the interface and residues distal from the interface. These mutations and the pulldown with PTPRK are described in the text and are included as additional panels to a modified Figure 3. All mutations have the expected effect on the interaction based on the predicted complex structure. To help illustrate the positions of these mutations we have also included a figure of the interface with the residues highlighted.

For the PTPRK-D2 we have also introduced two new mutations, one buried in the interface (F1225A) and one on the edge of the interface encompassing a loop that is different in PTPRM (labelled the M-loop). GST-Afadin WT protein was bound to GSH beads and tested for their ability to pulldown WT and mutated PTPRK. These new mutations (illustrated in the new Figure 4 —figure supplement 2) further support the model prediction. F1225A almost completely abolishes binding as predicted, while the M-loop retains binding. These mutations and their effects are now described in the main text and the pull-down data, including controls and retesting of the original DM mutant, are included as panel H in a newly modified Figure 4 focussed solely on the PTPRK interface.

2. Further characterization of the pY1230 antibodies to demonstrate their specificity for pY1230 Afadin.

We would like this antibody to be a useful and freely accessible tool in the field and have taken on board the request for additional validation. To this end we have significantly expanded Supplementary Figure 2 (now Figure 1 —figure supplement 2) and included a dedicated section of the results as follows:

1. We have now included information about all of the Afadin antibodies used in this study, since Afadin(BD) appears to be sensitive to phosphorylation (Figure 1 —figure supplement 2A).

2. We have demonstrated that the Afadin pY1230 antibody detects an upregulated band in PTPRK KO MCF10A cells, consistent with our previous tyrosine phosphoproteomics (Figure 1 —figure supplement 2B). This indicates that the antibody can be used to detect endogenous Afadin phosphorylation.

3. We have included two new knock down experiments demonstrating the recognition of Afadin by our antibody (Figure 1 —figure supplement 2C). There appear to be two Afadin isoforms recognised in HEK293T cells by both the BD and pY1230 antibody, consistent with previous reports (Umeda et al. MBoC, 2015). We have highlighted these in the figure.

4. We have performed mutagenesis to demonstrate the specificity of the antibody. We tagged Afadin with a fluorescent protein tag, reasoning that it would cause a shift in molecular weight that could be resolved by SDS PAGE, as is the case. We noted that the phosphopeptide used spans an additional tyrosine, Y1226, which has been detected as phosphorylated (although to a much lower extent than Y1230) on Phosphosite plus. The data clearly show that Afadin cannot be phosphorylated when Y1230 is mutated to a phenylalanine (compared to CIP control), indicating that this is the predominant site recognised by the antibody. In addition, the endogenous pervanadate-stimulated signal is completely abolished by CIP treatment (Figure 1 —figure supplement 2D).

5. We have included densitometric quantification of the dephosphorylation assay shown in Figure 1B, which was part of a time course and shows preferential dephosphorylation by the PTPRK ICD compared to the PTPRK D1. The signal stops declining with time, which could indicate antibody background, or an inaccessible pool of Afadin-pY1230 (Figure 1 —figure supplement 2E).

6. To further demonstrate that this site is modulated by PTPRK in post-confluent cells, we have used doxycycline (dox)-inducible cell lines generated in Fearnley et al., 2019. Upon treatment with 500 ng/ml Dox for 48 hours PTPRK is induced to lower levels than wildtype, however, normalized quantification of the Afadin pY1230 against the Afadin (CST) signal clearly indicates downregulation by PTPRK WT, but not the catalytically inactive mutant (Figure 1 —figure supplement 2F and 2G).

Together these data strengthen our assertion that this antibody recognises endogenously phosphorylated Afadin at site Y1230, which is modulated in vitro and in cells by PTPRK phosphatase activity. For clarity, we have highlighted and annotated the relevant bands in figures. We have also included identifiers for each Afadin total antibody was used in particular experiments.

Some additional suggestions for improvement can be found in the individual reviews, where one particular concern is that the anti-pY1230 Afadin antibodies have not been sufficiently well characterized. We look forward to receiving a revised version of your paper describing this exciting new mechanism in which the poorly understood pseudophosphatase domain in PTPRK acts to recruit pTyr substrate proteins.Reviewer #1 (Recommendations for the authors):1. The AF2 model of the Afadin 1393-1455 region bound to the PTPRK D2 domain is supported by the properties of the acidic pocket DM D2 mutant, but, in the absence of a co-crystal structure, it would be reassuring if some additional mutations in the D2 domain as well as in the Afadin core binding motif were analyzed.

This request is encompassed by essential revisions point 1.

2. The DM mutant data show that the D2 domain interaction is important for Afadin pTyr dephosphorylation in vitro, but one would also like evidence that the DM PTPRK mutant lacks Afadin pY1230 dephosphorylating activity in cells. The authors have the PTPRK KO MCF10A cells they generated in their first eLife paper that could be used to re-express the WT and DM PTPRK and then monitor Afadin dephosphorylation with their new anti-pY1230 antibodies.

We agree with the reviewer that this would be an interesting experiment. Although we do have the PTPRK KO line these experiments would require careful control of PTPRK mutant expression levels therefore needing generation of new stable cell lines. This work will require substantial time and resources delaying the publication of this current work. As this has not been requested as an essential revision we respectfully acknowledge that these would be excellent experiments for future work.

3. From the model structure in Figure 3B, it is not clear whether the Afadin helix binds to the backside or frontside of the D2 domain. In other words, where is the Afadin binding pocket relative to the "active" site of D2, and would oxidation of the active cite Cys be expected to alter Afadin binding? Some further discussion is called for.

We have made a series of additional figures (included as a panel in new Figure 4A) to clearly illustrate that the D2 “active” site is on the opposite face from the Afadin binding site. Due to the localisation of the “active” site so far from the binding face we would not expect oxidation of this cysteine to alter binding. All experiments were done in the presence of DTT or TCEP, representing binding under reducing conditions.

4. The blotting data for the Afadin band in Figures 5F and G are not very clean and need quantifying. In the experiment in panel F it is not clear why the authors did not immunoblot an Afadin IP with their new pY1230 antibodies, instead of precipitating with anti-pTyr antibodies and then blotting with anti-Afadin antibodies. This would enable them to demonstrate that the Afadin/D2 interaction specifically promotes dephosphorylation of pY1230 as opposed to dephosphorylation of other pTyr residues in Afadin (admittedly panel G suggests that pY1230 is not dephosphorylated by the DM mutant). With regard to these experiments, the authors should indicate in the text that the pervanadate in the lysate was neutralized by addition of an excess of DTT (presumably before addition of the recombinant PTPRK proteins).

We carried out our dephosphorylation assays on pTyr IPs since this allows us to include multiple controls, such as p120-catenin, which is equally dephosphorylated by WT PTPRK and the DM mutant. If we had IP’d Afadin, we would also have had to IP p120-catenin and paxillin as controls. We agree that the DM mutant does not dephosphorylate pY1230 (now quantified in new Figure 5 —figure supplement 1) when measured directly. The DM mutant clearly impairs the ability of PTPRK to dephosphorylate Afadin.

We have now indicated in the Results section that DTT is used to quench any excess pervanadate.

5. The anti-pY1230 Afadin antibodies should be more thoroughly characterized for their specificity. For instance, based on the top panel of Figure S2, it is not clear what band the reader is supposed to focus on – presumably the upper band but the "pY1230" signal in this band did not go down as much as the Afadin protein band in the second panel in the Afadin siRNA sample. The authors need to use a Y1230F mutant Afadin and YopH PTP treatment of the pV lysate as a negative controls, etc., to validate the specificity of these antibodies. If the new pY1230 antibodies can be shown to be specific, it would be interesting to know whether they can be used to detect pY1230 phosphorylated Afadin by IF, and whether it colocalizes with PTPRK.

We have included additional characterisation of the phosphoantibody addressed in the response to essential revisions point 2.

As suggested, we tested the antibody in immunofluorescence experiments, but did not detect a signal that could be modulated by EGF treatment. This may require further optimisation, or may not be a viable application for the antibody.

Reviewer #2 (Recommendations for the authors):1. Would recommend reporting in the figure legend how many experimental replicates have been collected for each experiment.

Where appropriate we have added the number of replicates for experiments. Several of the pulldowns were done using different combinations of constructs but all were tested a minimum of three times therefore we have stated that gels shown are “representative of n ≥ 3 independent experiments”. We have now included time course quantifications for dephosphorylation assays. For cell line experiments, replicates are either shown explicitly or reported in figure legends.

2. Related to #1, it is noted that the mean and SEM in Figure 2E are based on only two replicates.

ITC experiments require very large quantities of highly pure protein samples making them extremely resource intensive. Multiple measurements are made throughout the titration meaning that each single experiment involves 12 data points that are fit to a single binding model. We carried out the experiment twice with proteins purified independently for each experiment and obtained excellent agreement for the calculated affinity. We therefore did not consider additional ITC experiments to be necessary, especially in light of their resource-intensive nature.

3. The statement that the binding of the anti-afadin antibody is affected by phosphorylation could be better supported by experiments via transfection of tagged afadin followed by blotting with anti-tag vs anti-afadin antibodies.

Please see response to essential revisions point 2.

4. Is the "reverse" mutation of PTPRM residues homolog to PTPRK aa 1273 and 1335 respectively into H and L sufficient to rescue the interaction with afadin and its dephosphorylation by PTPRM?

Although well-designed single point mutations can disrupt a protein-protein interaction it is very rare to identify a gain-of-function mutation as it is not truly a “reverse” experiment. An interaction interface is the sum of many individual interactions that all contribute to the overall binding strength. Dropping the affinity below a certain threshold is relatively straight forward if the correct mutation is selected. However, introducing a mutation that can, on its own, mediate a new interaction is much less likely. As this experiment was not included as an essential revision we respectfully decline to mutate PTPRM into a more PTPRK-like molecule for this manuscript. To further support that it is the protein-protein interaction, and not the phosphosite, that is key to specificity, we have previously demonstrated that a chimeric recombinant PTPRM-D1-PTPRK-D2 protein is competent to dephosphorylate Afadin (Fearnley et al. 2019).

5. Could the authors check the numbering of residues in Supplementary Figure 9: residue 1273 appears to be an L in that figure.

We thank the reviewer for pointing out this mistake, it is now corrected. Note this figure is now labelled Figure 4 —figure supplement 2.